# Site-selective superassembly of biomimetic nanorobots enabling deep penetration into tumor with stiff stroma

Miao Yan[1,11], Qing Chen[2,11], Tianyi Liu[1], Xiaofeng Li[3], Peng Pei[1], Lei Zhou[2], Shan Zhou[1], Runhao Zhang[1], Kang Liang[4], Jian Dong[2], Xunbin Wei[5], Jinqiang Wang[6], Osamu Terasaki[7], Pu Chen[8], Zhen Gu[6], Libo Jiang[2]✉ & Biao Kong[1,9,10]✉

Chemotherapy remains as the first-choice treatment option for triple-negative breast cancer (TNBC). However, the limited tumor penetration and low cellular internalization efficiency of current nanocarrier-based systems impede the access of anticancer drugs to TNBC with dense stroma and thereby greatly restricts clinical therapeutic efficacy, especially for TNBC bone metastasis. In this work, biomimetic head/hollow tail nanorobots were designed through a site-selective superassembly strategy. We show that nanorobots enable efficient remodeling of the dense tumor stromal microenvironments (TSM) for deep tumor penetration. Furthermore, the self-movement ability and spiky head markedly promote interfacial cellular uptake efficacy, transvascular extravasation, and intratumoral penetration. These nanorobots, which integrate deep tumor penetration, active cellular internalization, near-infrared (NIR) light-responsive release, and photothermal therapy capacities into a single nanodevice efficiently suppress tumor growth in a bone metastasis female mouse model of TNBC and also demonstrate potent antitumor efficacy in three different subcutaneous tumor models.

Anticancer nanomedicines as an emerging therapeutic platform have the potential to revolutionize the treatment of cancer[1–3]. However, limited tumor penetration remains major bottlenecks for the delivery of nanomedicines into tumors, thus greatly hampered clinical translatability, especially for triple-negative breast cancer (TNBC). TNBC has negative expression of common breast cancer antigens including estrogen receptor (ER), progesterone receptor (PR), and human epidermal growth factor receptor 2 (HER2)[4]; therefore, TNBC and its metastasis show no response to endocrine or targeted therapies, making chemotherapy the first-choice treatment option[5–7]. However, the highly fibrotic, overly connective tissue hyperplasia and a viscous extracellular matrix (ECM) in TNBC naturally constitute a particularly dense physical barrier, greatly hindering access to tumor, let alone entry into cell for subsequent anticancer drug delivery[7,8]. Considerable

[1]Department of Chemistry, Shanghai Key Lab of Molecular Catalysis and Innovative Materials, iChEM, Fudan University, 200438 Shanghai, P. R. China. [2]Department of Orthopaedic Surgery, Zhongshan Hospital, Fudan University, No. 180 Fenglin Road, 200032 Shanghai, P. R. China. [3]Department of Chemistry, The University of Hong Kong, Hong Kong 999077, P. R. China. [4]School of Chemical Engineering and Graduate School of Biomedical Engineering, The University of New South Wales, Sydney, NSW 2052, Australia. [5]Biomedical Engineering Department and Cancer Hospital and Institute, Key Laboratory of Carcinogenesis and Translational Research, Peking University, 100081 Beijing, P. R. China. [6]Key Laboratory of Advanced Drug Delivery Systems of Zhejiang Province, College of Pharmaceutical Sciences, Zhejiang University, 310063 Hangzhou, P. R. China. [7]School of Physical Science and Technology, ShanghaiTech University, 201210 Shanghai, P. R. China. [8]Department of Chemical Engineering, University of Waterloo, Waterloo, ON N2L 3G1, Canada. [9]Yiwu Research Institute of Fudan University, 322000 Yiwu, Zhejiang, P. R. China. [10]Shandong Research Institute, Fudan University, 250103 Jinan, Shandong, P. R. China. [11]These authors contributed equally: Miao Yan, Qing Chen. ✉e-mail: jiang.libo@zs-hospital.sh.cn; bkong@fudan.edu.cn

efforts have been made to improve tumor penetration by modulating nanomedicine shape[9,10], size[11], charge[5], aspect ratio[12,13], surface chemistry, and rigidity[14]; however, the efficiency remains limited due to the unchanged passive diffusion of bulky nanomedicines[2]. One key challenge in this field is, therefore, how to overcome biological barriers to improve nanomedicine translation and exploitation.

Self-propelling micro/nanorobots can improve tumor penetration due to active motion capability, beyond the ordinary diffusion limits[15,16], and therefore have drawn tremendous attention over 15 years. However, when applied in vivo to penetrate tumors with stiff stroma, the dense stroma could still impede the nanorobot motion. Furthermore, existing nanorobots lack strong binding to the cell membrane to promote engulfment; thus, nanorobots are unable to release drugs directly into the cytosol. Multifarious surfaces with nanotopography, such as those of nanostars, nanopollens, and virus-like and urchin-like nanoparticles, have been recognized as powerful morphologies strongly influencing membrane binding and destabilization for efficient cellular internalization[13,17–22]. Unfortunately, nanorobots anchored with nanotopological surfaces have rarely been reported. To address these issues using nanotechnology, it is essential to combine the rational design of nanocarriers with the fundamental understanding of the tumor biology.

In addition, efficient drug-loading capacity and controllable delivery should also be considered for an improved therapeutic index[23,24]. Hollow nanocarriers have been widely used for drug delivery because of their high surface areas, large cavity volume, and efficient loading capacity. In general, thermoresponsive materials and drugs are loaded inside the hollow cavities for constructing smart release system[25,26]. When subjected to heating, the thermoresponsive materials can be melted and the drugs are released through an opening. During this process, photothermal agents are required to trigger temperature changes. The most widely employed photothermal agents are organic near-infrared (NIR) dye molecules. In comparison with organic NIR dyes, the inorganic noble nanoparticles have stronger NIR absorption, high photothermal conversion efficiency, limited normal tissue damage, and high stability[23,25]. However, the synthetic capabilities of hollow nanocarriers are still largely limited to single component and enclosed morphology, not to mention more sophisticated hollow system equipped with an opening for efficient drug-carrying capability and controllable delivery.

Here, we show a site-selective superassembly strategy to synthesize a class of nanorobots that synergistically combines multiple features of advanced nanomedicines to address these issues. Each nanorobot consists of a AuNS half-coated with a thin SiO$_2$ shell in the head region and an open hollow tail connected to the half-shell. The site-selective superassembly growth of silica shell on the rough surface of AuNSs can be well controlled with a variable surface spiky length (~13–94 nm), tail length (~0–510 nm), and large tunable hollow diameter (~100–240 nm). By precisely controlling the ligand coverage location, silica can completely inherit the surface spiky topological structures, creating a protruding spiky head. As a result, the biomimic spiky surface nanotopologies and active motion capability markedly promote cellular uptake efficacy, transvascular extravasation, and intratumoral penetration. The excellent photothermal effect of AuNSs remodels tumor stromal microenvironments (TSM) by reducing stromal cell viability and causing denaturation of extracellular matrix, which could open the intratumoral path space for nanorobot motion in vivo and significantly enhance accessibility to cancer cells. Furthermore, the large open hollow tail enables co-encapsulation of stimuli-responsive phase-change materials (PCMs) and doxorubicin (DOX) drugs, which can be triggered by 980 nm NIR irradiation due to the excellent photothermal effect of AuNS to raise the temperature beyond their melting point (~39 °C), realizing on-demand delivery. These rationally designed nanorobots exhibit prominent therapeutic effects, eliminating 87.78% of the tumor volume within 20 days in a

spinal bone metastasis model of TNBC in mice (Fig. 1), and also demonstrate similar functionality in pancreatic cancer, melanoma, and lung cancer-derived subcutaneous tumor models in mice.

## Results

### Design and characterization of the asymmetric urchin head/hollow tail nanostructures (UHHTNs)

The AuNSs were first added in a isopropyl alcohol (IPA)-H$_2$O mixture (2.5:1 V/V) containing (4-mercaptophenylacetic acid, 4-MPAA) and poly acrylic acid (PAA, M.W. = 5500), followed by stirring at room temperature for 30 min. The two types of ligands competition for surface binding sites enables the Janus like segregation on the surface of AuNSs[27], with one side is 4-MPAA region, and the other side is PAA domain. Then, hexadecyltrimethylammonium bromide (CTAB), tetraethyl orthosilicate (TEOS) and ammonium hydroxide were added in sequence. The rich –COOH group of free PAA molecular chain in solution interacted with the NH$_4^+$ to form PAA–NH$_4$ complex as a salt[28,29], leading to the mixture phase separation, with one enriches water and salt, the other one is mostly IPA, and thus a large number of small droplets are generated[30,31].

Silica was first deposited on the 4-MPAA side to form half-coated thin SiO$_2$ layer as the carboxylic groups of 4-MPAA could react with the silane monomer (Supplementary Fig. 1), and at PAA side, PAA polymer can be like a 'sponge' to absorb and reserve droplets inside its net structure due to their superhydrophilic nature, enabling continuous directional fusion of the droplets extracted from solution upon collision, which has been studied well in our recent work[32]. The CTAB molecules could adsorb onto the surface of PAA via Coulomb force and electrostatic interaction between CTA$^+$ and PAA$^-$. Due to the charge interaction, the silica oligomers interacted with CTA$^+$. Continuous the small droplet fusion onto PAA along longitude enable silica continuous deposition at the interface between PAA-CTAB to form an open hollow tail connected to the half shell of 4-MPAA-covered surface. The self-nucleated hollow silica nanoparticles formed in solution further confirmed the generation of free droplets (Supplementary Fig. 2). If only 4-MPAA was used, the 4-MPAA covered the entire surface of the AuNSs and no small droplets formed, resulting in a fully encapsulated AuNS@silica core-shell structure (Supplementary Fig. 3).

Asymmetric urchin head/hollow tail nanostructures (UHHTNs) consist of a NIR-absorbed AuNS half-coated with a SiO$_2$ shell in the head region and an open hollow tail connected to the half shell (Fig. 2a). The employed AuNSs have nanospikes and an average diameter of ~265 nm (Fig. 2b, Supplementary Fig. 4). Transmission electron microscopy (TEM) images revealed the uniform morphology of the as-synthesized UHHTNs with high yield (almost 100%). High-resolution transmission electron microscopy and selected area electron diffraction results show that the SiO$_2$ shell in the hollow tail is amorphous (Fig. 2c). UHHTNs have a spiky head with a width of ~285 nm and a hollow tail with a dedicated opening of ~100 nm at the end of the hollow tail (Fig. 2d). The average body length of UHHTN is ~351 nm (Fig. 2e). Scanning electron microscopy (SEM) image reveals that there are many nanospikes on the surface of the UHHTNs that resemble the surface spiky morphology of some urchin (Fig. 2f). The thicknesses of silica in the head and tail regions marked with blue rectangular frames are ~6.78 ± 1.21 nm and 9.06 ± 1.52 nm, respectively (Supplementary Fig. 5). The outlines of the SiO$_2$ and AuNS compartments can be clearly detected by elemental mapping and matched well with the relative positions in the UHHTN (Fig. 2g).

The UV–vis–NIR absorption spectra of as-prepared AuNSs and UHHTNs were performed. The bare AuNSs have a broad absorption range from 800 to 1200 nm and an intense absorption at ~916 nm in the NIR region, resulting from their localized surface plasmon resonance (LSPR). After the SiO$_2$ coating, the peak was redshifted by 23 nm from 916 to 940 nm due to an increase in the refractive index of the medium surrounding the AuNSs. Note that before and after the SiO$_2$

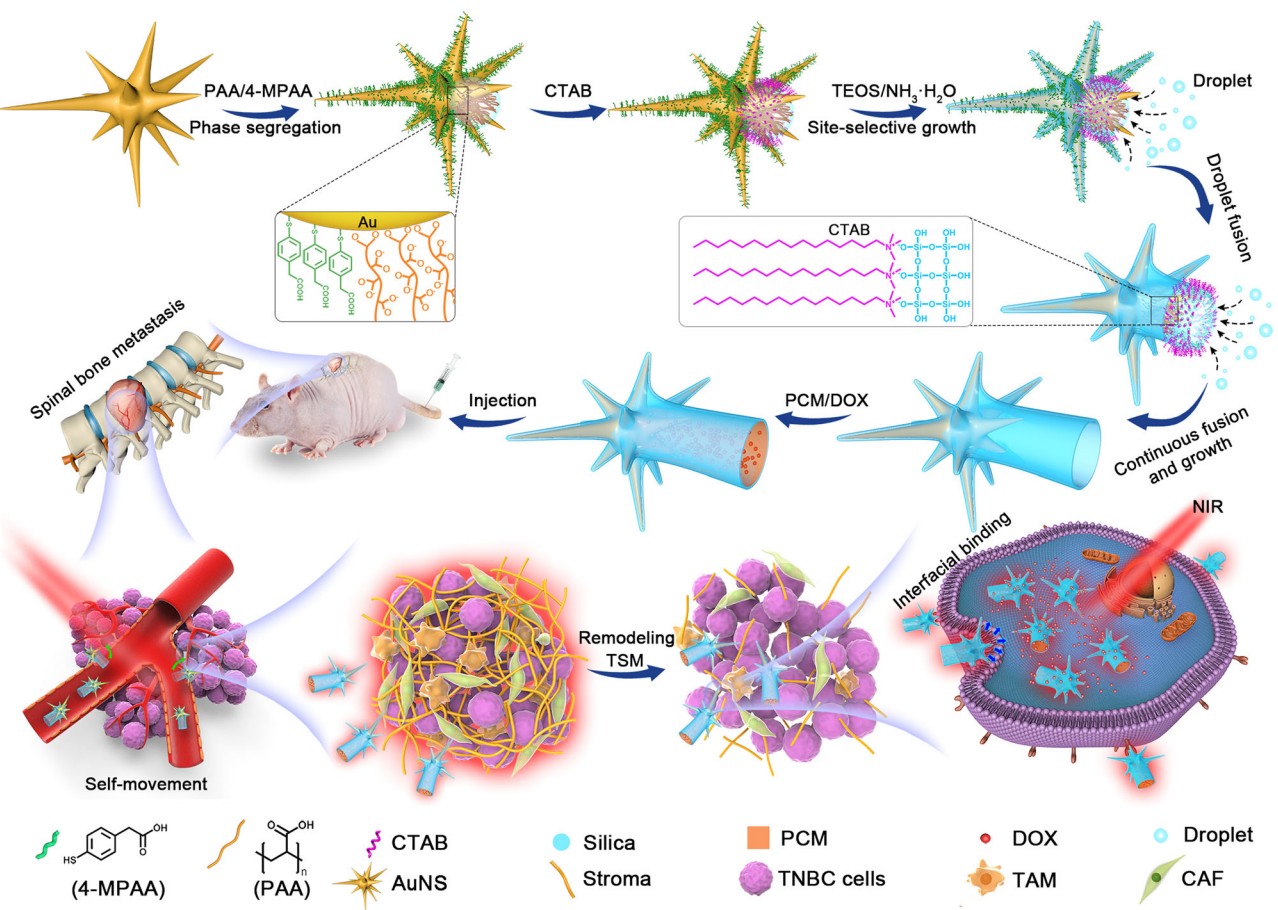

**Fig. 1 | Design of the superassembled nanorobot for the treatment of TNBC spinal bone metastasis with stiff stroma.** Schematic illustration of the site-selective superassemble process of UHHTN nanorobots and the process of TNBC treatment with UHHTN nanorobots mainly including transvascular penetration, TSM remodeling, deep tumor penetration, and active cellular internalization.

coating, no appreciable broadening of the absorption spectrum was occurred, indicating that AuNSs did not form aggregates during the growth process (Fig. 2h). The dedicated architecture still remained intact even after long periods of ultrasonication (Supplementary Fig. 6). Moreover, the UHHTN aqueous solution showed obvious light scattering capacity after storage of 2 months and then stirring, while large precipitates were formed for bare AuNS aqueous solution, indicating that the UHHTNs possess good dispersion stability with the protection of thin silica shell (Fig. 2i). This approach easily produced large-scale monodisperse UHHTN colloids (Supplementary Fig. 7).

**Tuning of the surface spiky topologies and hollow tail diameters**
The surface nanotopologies of nanostructures can be continuously varied by changing only the 4-MPAA concentrations. When the amount of 4-MPAA added is 5 mM, only nanostructures with relatively smooth surfaces (spiky length of ~13 nm) in the head region were obtained (Fig. 3a (left), b, e and supplementary Fig. 8). In the case of 15 mM, the spiky length on the surface was increased to 65 nm (Fig. 3a (middle), c, f and Supplementary Fig. 9) and simultaneously spike number increases. A further increase in the 4-MPAA concentration led to a much more and longer spike with a length of ~94 nm (Fig. 3a (right), d, g and Supplementary Fig. 10). The cavity diameters in b, c, and d are ~240 nm, 115 nm and 100 nm (indicated by yellow line), respectively.

Moreover, it can be noted that the spike lengths increase as the 4-MPAA concentration increases, while hollow tail diameters decrease. The corresponding hollow diameter of the tail decreased from ~240 nm to 115 nm to 100 nm (Fig. 3h). In addition, the resultant hollow tail has varied length, which are mainly

attributed different growth rates (Fig. 3i). The smaller the tail diameter was, the faster the growth rate became, and therefore the resulting tails had varied lengths.

Based on the above observations, a convincing mechanism was proposed to explain the formation of spiky surface in head. It has been demonstrated that specific binding locations of surface ligands can produce "active growth sites" and "inactivated growth sites", leading to specific site-selective growth[33–35]. In our system, nanospike length of UNNTH is closely related to silica deposition site in head region, which is determined by 4-MPAA coverage location because the carboxylic groups of 4-MPAA could react with the silane monomer as mentioned above.

A low 4-MPAA concentration cannot ensure that 4-MPAA molecules can fully cover entire site in the 4-MPAA region of AuNS, thus the 4-MPAA molecules preferentially adsorbed on the high-curvature sites of AuNSs such as the vertices due to their high surface energy, and further crosslinked into a thin silica layer along all vertices of 4-MPAA-covered AuNSs (Fig. 3j). Smooth surfaces without nanospikes were obtained in the head area. The intermediates exhibited a large cavity length in the head region (Supplementary Fig. 11) with no silica in the groove, confirming that silica growth did not occur in the groove region.

As the concentration increases, the cavity length in the head was significantly reduced (Supplementary Fig. 12), implying that the 4-MPAA started to appear at the edges and caused the silica to gradually deposit inward (Fig. 3k). When enough amount of 4-MPAA was added, the 4-MPAA molecules can sufficiently cover on the vertices and edges. In this case, silica growth inherited the surface topology

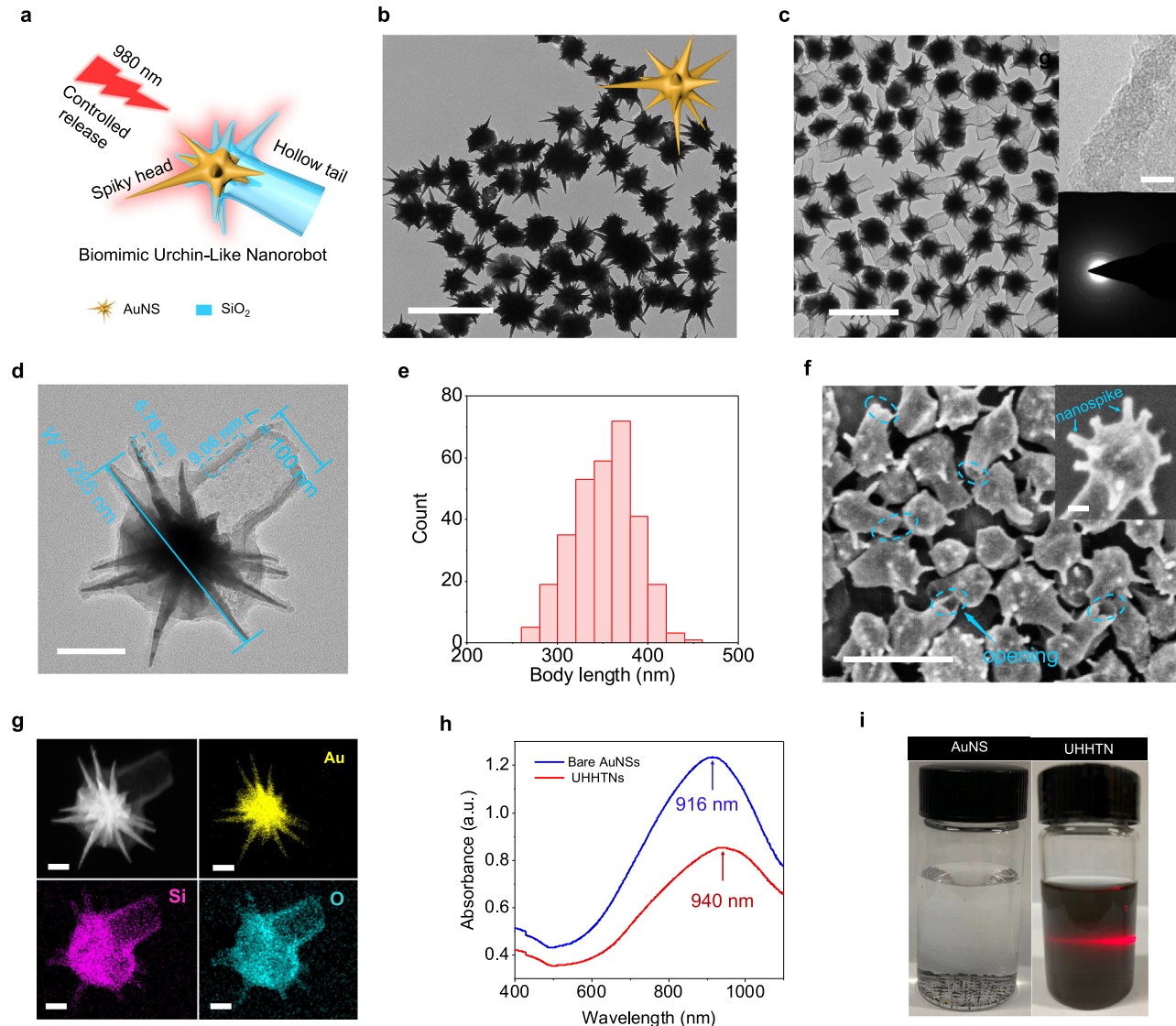

**Fig. 2 | UHHTNs fabrication and characterization. a** Schematic representation of UHHTN with NIR-activated photothermal property generated in AuNS. **b** TEM images of AuNSs and inset shows a 3D structural model of the AuNS. **c** Low-resolution TEM image of the as-synthesized UHHTNs. Insets: a high-resolution TEM image of SiO$_2$ tail wall (top right) and the corresponding electron diffraction pattern (bottom right). **d** Magnified TEM image showing an individual UHHTN with sharp surface spikes and an opening of ~100 nm at the end of the hollow tail ($L_{opening}$), a head width ($W_{head}$) of ~285 nm (indicated by blue line). **e** Body length distribution of UHHTN determined by TEM (350 UHHTNs analyzed). **f** SEM image. **g** elemental mapping of the UHHTNs. **h** UV-vis-NIR spectra recorded from aqueous suspensions of the as-obtained AuNSs (blue line) and UHHTNs (red line). Scale bars are 500 nm in (**b, c, f**) 5 nm in the inset of **c**, 100 nm in (**d**), and 50 nm in the inset of (**f, g**). **i** Photograph of the AuNS (left) and UHHTN (right) aqueous solution. Experiment was repeated three times independently with similar results.

from AuNSs, the cavity in the head almost disappeared (Supplementary Fig. 13) and eventually developed into the distinctive spiky surface (Fig. 3l). For the variation of hollow tail diameter, it can be explained by ligand competition theory that 4-MPAA concentration not only affect coverage degree, but also affects the two types of ligand occupied area[27]. Hence, an increased amount of 4-MPAA enables complete coverage on the vertices and edges, and simultaneously results in a smaller PAA region on the AuNS surface. As a result, the surface spiky length gradually becomes longer, whereas the tail diameter gradually becomes smaller. These results indicated that the ligand concentration allows fine control over the spiky surface topologies and hollow diameter.

## Thermo-responsive drug delivery and motion performance of UHHTN nanorobots

AuNSs with sharp spikes have gained considerable attention as photothermal agents because of their excellent photothermal properties.

For this reason, the optical properties of UHHTNs were studied carefully. UHHTNs with more and longer nanospikes (spike length of ~94 nm and body length of ~351 nm shown in Fig. 2c) were chosen as a model structure. The temperature of the UHHTN aqueous solution (the concentration was 100 µg/mL) increased to ~40, 48, 55, and 62 °C under 980 nm laser irradiation at varied power densities of 0.5, 1.0, 1.5, 2.0 W cm$^{-2}$ for 10 min, respectively (Fig. 4a), while that of the pure water increased only to 35.1 °C (Supplementary Fig. 14). The modified thin silica shell body with an opening does not shield the photothermal conversion performance of the AuNS core (Supplementary Fig. 15).

We then encapsulated fatty acids into the UHHTNs through the opening of the hollow tails. Briefly, UHHTNs were dispersed in a eutectic mixture of lauric acid (melting point (m.p.): 44 °C) and stearic acid (m.p.: 69 °C) at a weight ratio of 4:1. The loaded UHHTNs (denoted as PCM-UHHTN) were retrieved by centrifugation, followed by gentle washing with DMSO to remove the surface adsorption and free fatty acids. Then water was added to solidify the DOX-trapped

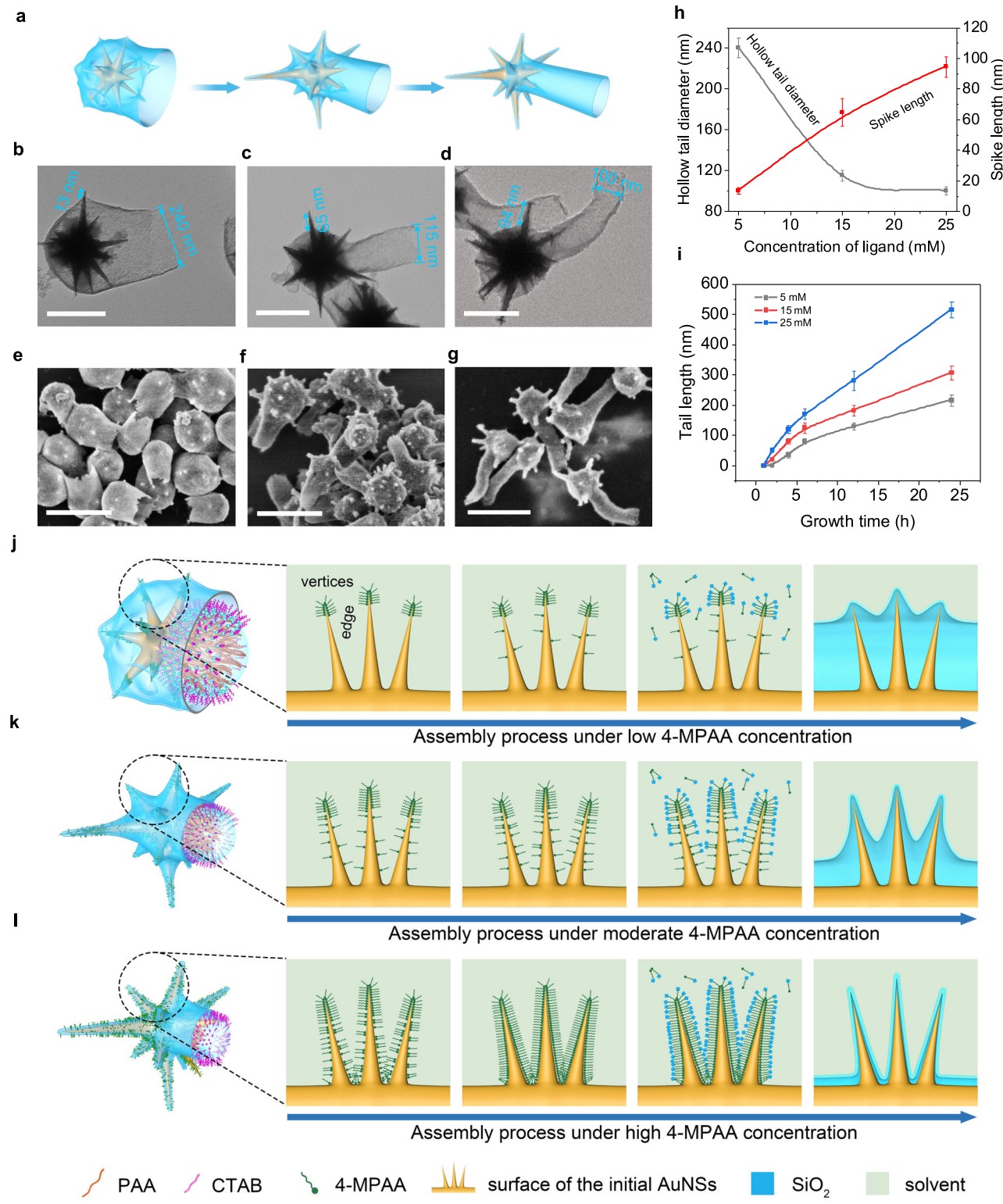

**Fig. 3 | Formation mechanism of the spiky topology. a** Schematics illustrating the nanostructures with different surface spiky topologies and hollow tail diameters. **b–d** TEM and (**e–g**) SEM images of nanostructures prepared at various 4-MPAA concentrations: 5 mM (**b, e**), 15 mM (**c, f**), and 25 mM (**d, g**). Scale bars are 200 nm in (**b–d**) and 500 nm in (**e–g**). **h** Hollow tail diameter and corresponding spike length of nanostructures prepared in different 4-MPAA concentrations. The data are shown as the mean ± s.d. **i** Tail length at different 4-MPAA concentrations at different time points are shown. The data are shown as the mean ± s.d. Experimental data in (**h** and **i**) are mean ± s.d. of samples in a representative experiment (*n* = 50). **j–l** Schematic illustration of site-selective superassembly of silica on the 4-MPAA-covered surface of AuNSs at (**j**) low, (**k**) moderate and (**l**) high 4-MPAA concentrations. Source data for the figure are provided within the Source Data file.

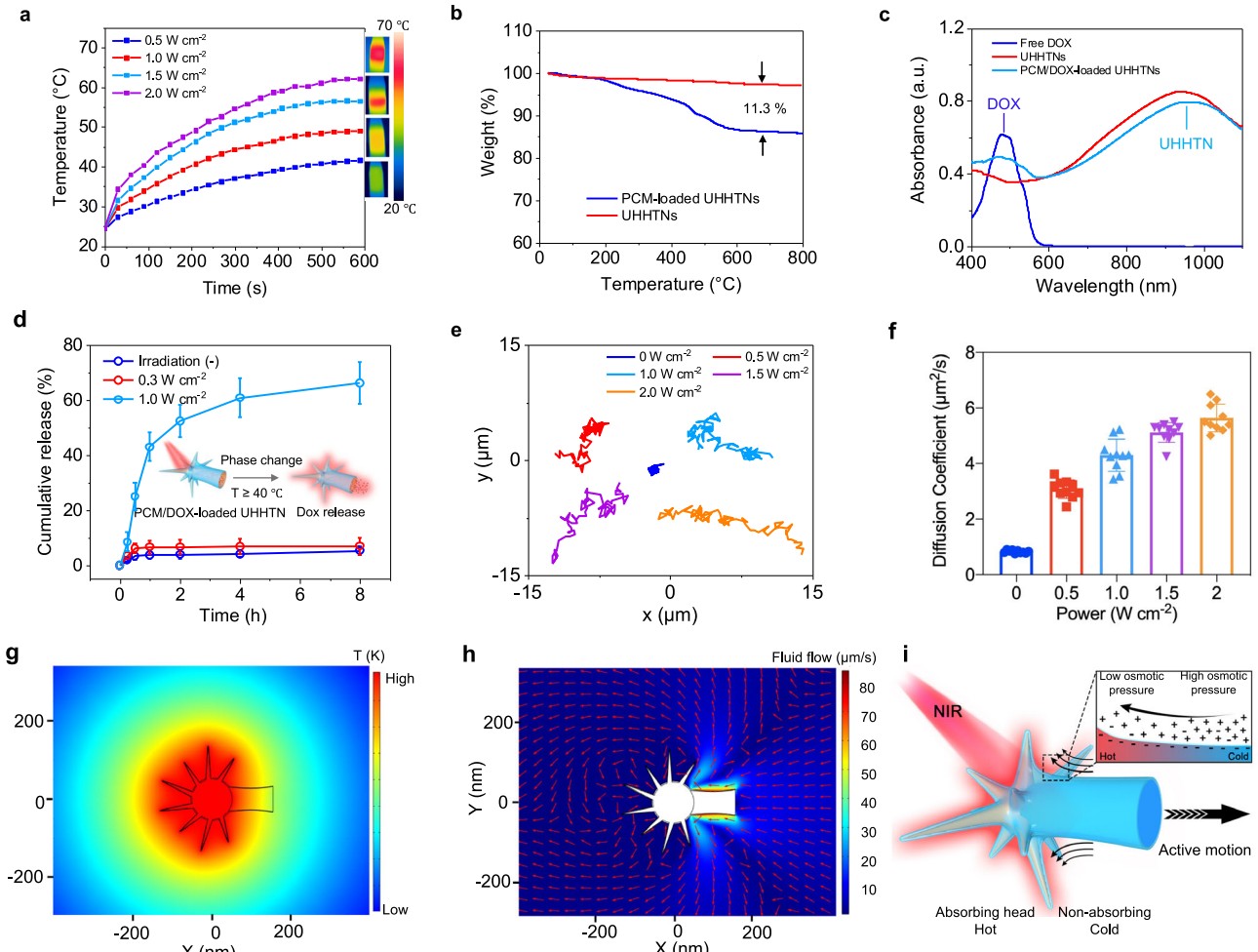

**Fig. 4 | Photothermal properties, thermo-responsive drug delivery, and motion evaluation of nanorobots. a** Photothermal effect of UHHTNs in aqueous solution upon 980 nm laser irradiation under different power densities for 10 min. **b** TGA data obtained from the pristine UHHTNs (red) and UHHTNs containing PCM (blue). The 11.3% weight loss corresponds to the evaporation of PCM inside UHHTNs. **c** UV-vis-NIR spectra recorded from an aqueous suspension from free DOX (blue line), UHHTNs (red line) and UHHTNs loaded with PCM-DOX (green line). **d** Drug release profiles of DOX from the UHHTNs in DMEM with 10% serum under NIR light sti- mulation with different laser power densities. Inset: Schematic representation of the thermo-responsive drug delivery. Data are presented as mean ± s.d ($n = 3$

replicates). **e** Trajectories of PCM/nanorobots irradiated with different NIR power densities for 10 s (Supplementary Videos 1–5). **f** The average diffusion coefficient ($D$) values. Experimental data are mean ± s.d. of samples in a representative experiment ($n = 10$). **g** Steady-state temperature distribution of a PCM/nanorobot under NIR irradiation. **h** fluid velocity profile. **i** Schematic showing a mechanism of thermophoresis. When NIR light is locally absorbed on one side of a UHHTN, a local temperature gradient is formed across the UHHTN surface. This temperature gradient results in a corresponding osmotic pressure gradient which induces fluid flow at the UHHTN solvent interface. Source data for the figure are provided within the Source Data file.

fatty acids inside the cavity. Owing to their excellent biocompatibility and biodegradability, these two fatty acids can be safely used in bio- medicine. The differential scanning calorimetry (DSC) curve exhibited a sharp melting point at ~40 °C, which is slightly higher than the normal temperature of the human body (37 °C) (Supplementary Fig. 16). The thermal gravimetric analysis (TGA) of the PCM-UHHTNs showed a loading amount of 11.3 wt% for PCM, indicating the PCM can be loaded into the cavity through the opening (Fig. 4b). Furthermore, the PCM and DOX were co-encapsulated into the cavity for NIR-triggered drug release (Supplementary Fig. 17). The UV−vis−NIR spectra and fluores- cence images confirmed the successful encapsulation of the DOX (Fig. 4c and Supplementary Fig. 18). TEM results further showed that PCM/DOX was loaded into the cavity of UHHTNs and no drug-loaded fatty acid particles are observed, implying that the final product is only PCM/DOX-trapped UHHTNs (Supplementary Fig. 19). The loading amount was calculated to be 110 mg per gram of the UHHTNs (Sup- plementary Fig. 20).

The release of encapsulated DOX could be readily tuned by varying the power density and/or the duration of laser irradiation. The

amount of released DOX reached 61% at a 1 W cm⁻² power density within 8 h, corresponding to an equilibrium temperature of 45 °C (Fig. 4d). In contrast, less than 3% and 5% of DOX were released at 0.3 W cm⁻² and 0 W cm⁻², respectively, indicating that the drug was retained inside the interior of the UHHTNs due to the limited diffusion of DOX molecules by the solid PCM.

We further explored the motion performance of PCM-UHHTNs. The trajectories of randomly selected nanorobots ($n = 10$) were tracked from the recorded videos by Image J under the condition of NIR irradiation with varied power densities and corresponding mean square displacement (MSD) were calculated (Fig. 4e and Supplemen- tary Fig. 21). The average MSD of nanorobot trajectories showed an increase with time (Supplementary Fig. 22). The effective diffusion coefficient ($D$) was obtained by applying the equation 1 and 2 shown in Supplementary Information. The diffusion coefficient of the UHHTN nanorobots increases from the Brownian diffusion value (~0.82 μm²/s) in the absence of NIR laser to 3.08 μm²/s at 0.5 W cm⁻² laser power, 4.28 μm²/s at 1.0 W cm⁻² laser power, 5.12 μm²/s at 1.5 W cm⁻² laser power, and 5.90 μm²/s at 2.0 W cm⁻² laser power (Fig. 4f). These results

confirmed that the movement of nanorobots can be facilely tuned by the incident NIR laser power.

To prove that the propulsion is attributed to asymmetry of UHHTNs, we use symmetric AuNS@SiO$_2$ core-shell nanoparticles with similar size and NIR-responsive photothermal effect for comparison (Supplementary Fig. 23). The trajectory shows random motion within short distance, suggesting that the motion is indeed morphology dependent (Supplementary Fig. 24). Based on these results, the motion of the nanorobot can be attributed to the generation of the local thermal gradients across the PCM/nanorobot, where the AuNSs in the head region convert the adsorbed photons to heat. The simulation results show that the temperature near the head region was higher than those near the tail opening (Fig. 4g and Supplementary Fig. 25). When a temperature gradient is formed along the interface between the solvent and the nanorobot, an osmotic pressure gradient parallel to the interface and antiparallel to the temperature gradient is created[36]. This is mainly due to the difference in ion concentration caused by the thermal gradient. The concentration is slightly higher at the cold side and thus the pressure is slightly higher at the cold side, and its gradient is opposite to thermal gradients. As a consequence, there is a thermoosmotic fluid flow along the surface toward a higher temperature[37]. The fluid velocity field profile shows that thermoosmotic flow on the outer surface of the UHHTNs drives the fluid to the hot end in the head (Fig. 4h). As the fluid is stationary within the laboratory frame of reference, this implies that the particle has to move along the temperature gradient opposite to the interfacial fluid flow and thus drives UHHTNs via a mechanism of thermophoresis (Fig. 4i).

### Cellular cytotoxicity, internalization, and in vitro penetration

To evaluate the effect of surface spikes and self-motile capability on cellular internalization, nanoparticles with smooth surfaces were synthesized for comparative purposes (Supplementary Figs. 26, 27). MDA-MB-231 cells were treated with 100 µg/mL DOX-loaded nanoparticles without nanospikes, spiky nanoparticles and spiky nanoparticles with laser irradiation for 0–30 min. Red fluorescence could already be found in the cytoplasm in the Spike group; however, the fluorescence was still scarce at 30 min in the No Spike group. When nanoparticles with spiky heads were explored under NIR irradiation (1 W cm$^{-2}$) for 0–30 min, the confocal results showed that more cells were stained with red fluorescence than those without laser irradiation (Fig. 5a and Supplementary Fig. 28). The cellular uptake efficiency was quantitatively evaluated by calculating the red fluorescence intensity (Fig. 5b). Similar results were demonstrated in three other cell lines, A549, PANC-1 and B16F10 (Fig. 5c–e; Supplementary Figs. 29–31). To confirm whether the UHHTN/DOX were internalized inside cellular interiors, the cell membranes of four cell lines were stained and the results show that more UHHTN/DOX nanorobots were located in the cytoplasm and nucleus (Supplementary Fig. 32). In addition, we also explored the internalization efficiency of UHHTN/DOX with different degrees of spikes shown in Fig. 3. The results show that the UHHTN/DOX with a much more and longer spike with a length of ~94 nm have higher internalization efficacy, which possibly attributed to more spikes with a certain length of spikes in nanoparticle surface enhance the nano-bio interfacial interaction (Supplementary Fig. 33).

Considering that the UHHTN could have a higher spike-mediated mechanical stress due to a smaller contact area between nanospikes and cell membrane, which could disrupt membrane to enter into interior of cells. We first confirm whether UHHTN/DOX entering inside cell is dependent on endocytosis, cells were incubated with the UHHTN/DOX at 4 °C, a temperature at which endocytosis is generally inhibited[38]. Our results show a certain decrease, suggesting that internalization is endocytosis-dependent. We further observed changes of cellular uptake behavior with specific endocytosis inhibitors. Pretreatment with mycoplasmin (fractal protein endocytosis

inhibitor), chlorpromazine (lattice protein endocytosis inhibitor), and monensin (lattice protein and fractal protein non-dependent endocytosis inhibitor) had no effect on UHHTN/DOX internalization, while very weak red fluorescent signals were visible around cells treated with cytochalasin D (macropinocytosis inhibitor), so we can infer that UHHTN/DOX internalization is mainly dependent on the macropinocytosis-mediated endocytosis pathway (Supplementary Fig. 34).

We further explore the distribution of UHHTN/DOX inside the cell. We labeled lysosomes and Golgi apparatus separately and observed the co-localization of nanorobots with both. Under NIR irradiation, more UHHTN/DOX entered the cell and were distributed in the Golgi apparatus, lysosomes, and nucleus (Supplementary Fig. 35). These results suggest that UHHTN/DOX can efficiently enter the intracellular compartment and promote nuclear delivery of DOX to exert antitumor effects.

To exclude the possibility that NIR irradiation (1 W cm$^{-2}$) may cause damage to the cells that could have an effect on cell internalization, CCK-8 analysis was performed and showed that the cell viability among these three groups after 0–30 min of incubation was similar (Supplementary Fig. 36). Furthermore, the cell viability remained greater than 90% under treatment with UHHTN/DOX for 24 and 48 h (Supplementary Fig. 37), demonstrating the cytocompatibility of UHHTN/DOX. Similar results were also observed in A549, PANC-1 and B16F10 cells (Supplementary Figs. 38).

Considering the transport of nanorobots in the body from blood vessels to specific tumor sites, we analyzed the transvascular and transcellular capacity. Two different models, including the penetration of vascular endothelial cells (endothelial cells in the upper chamber and tumor cells in the lower chamber) and tumor cells (tumor cells in both chambers) were used in a Transwell test (Fig. 5f). The result exhibited that large amount of UHHTN/DOX nanoparticles with laser irradiation can be transferred to MDA-MB-231 cells in the lower chamber than that without irradiation (Fig. 5g, left panel and Supplementary Fig. 39a). Similar results were shown in transcellular penetration (Fig. 5g, right panel and Supplementary Fig. 39b). Such penetration performance of the UHHTN/DOX nanorobots was also exhibited in the three constructed Transwell models of A549, PANC-1, and B16F10 cell lines, indicating that our nanorobot system function in a variety of tumors with stiff stroma (Supplementary Figs. 40–42).

In order to further investigate the penetration of UHHTN nanorobots, three-dimensional multicellular tumor spheroid (3D MTS) similar to solid tumors were established. It was found that more and higher-density red fluorescence throughout the 3D MTS was visible in Spike+NIR group compared with either Spike or No Spike group (Fig. 5h). The mean optical density (MOD) values were quantitative data on the red fluorescence signal generated by the three groups after penetration into the 3D MTS (Fig. 5i and Supplementary Fig. 43). The fluorescence intensity of the nanorobots in the equatorial plane was measured to be about 1.6 and 2.9 times higher than the other two groups, which suggests that nanorobots with the combination of active motility and nano-bio interaction can effectively promote their tumor penetration.

### Therapeutic efficacy and safety of UHHTN/DOX nanorobots

The therapeutic effect of UHHTN/DOX nanorobots in vitro was investigated. Considering the drug carrying capacity and photothermia effect of UHHTN/DOX nanorobots, the cytotoxicity of UHHTN/DOX nanorobots with 1.5 W cm$^{-2}$ laser irradiation for 30 min was confirmed by the live/dead cell analysis (Supplementary Fig. 44) and colony formation assay (Supplementary Fig. 45). Furthermore, the migration and invasion test of MDA-MB-231 cells demonstrated that UHHTN/DOX nanorobots exerted a powerful synergistic anti-TNBC cell efficacy, which might be ascribed to the enhanced cellular internalization of UHHTN nanorobots (Supplementary Figs. 46–48).

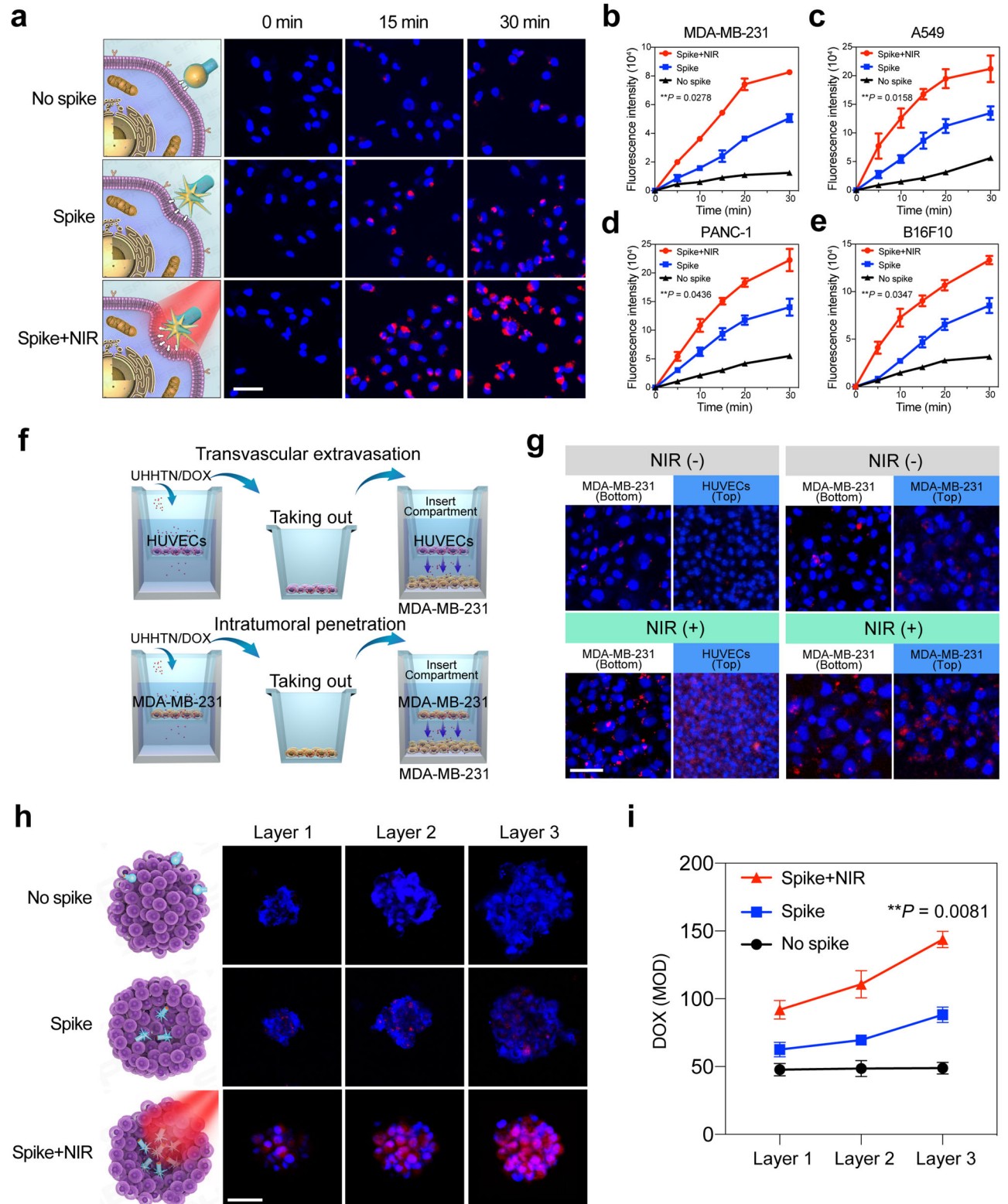

Spinal vertebral body is the main axial bone. It has been reported that spinal metastasis accounts for approximately 60% of TNBC distant bone metastasis with stiff stroma[39]; therefore, we selected a spinal metastasis model to analyze the biological functions of UHHTN nanorobots. To evaluate the in vivo therapeutic efficiency of UHHTN/DOX nanorobots for spinal metastasis, the mouse TNBC spinal metastasis models were established by intraspinal vertebral inoculation of MDA-MB-231 cells into mice, followed by intravenous nanoparticles injection and 1.5 W cm$^{-2}$ laser irradiation for 30 min. Fourteen

days after cell injection, these mice were randomly divided into 7 groups, which were intravenously given the following treatments: saline, DOX, Laser (980 nm laser with 1.5 W cm$^{-2}$ for 30 min), UHHTN, UHHTN+Laser, UHHTN/DOX, and UHHTN/DOX+Laser, followed by the bioluminescence imaging, survival status analysis, tumor measurement, histology, computed tomography (CT) and magnetic resonance imaging (MRI) examination at different stages (Fig. 6a). A 20-day independent experiment of survival recording was performed to observe the survival rate (Fig. 6b). The survival rates of nude mice

**Fig. 5 | In vitro penetration of UHHTN/DOX nanorobots. a** Fluorescence images of MDA-MB-231 cells treated with nanoparticles without nanospikes, spiky nanoparticles and spiky nanorobots after 0, 15 and 30 min of intervention. Blue fluorescence indicates the nucleus of the stained cell. Scale bar, 25 µm. **b**–**e** Internalized fluorescence density of nanoparticles without nanospikes, spiky nanoparticles and spiky nanorobots in (**b**) MDA-MB-231, (**c**) A549, (**d**) PANC-1 and (**e**) B16F10 tumor cells, respectively. Data are presented as the mean ± s.d. Statistical significance was determined by one-way ANOVA with Tukey's post hoc test for comparison. ($n = 3$ replicates; **$P < 0.05$, ***$P < 0.001$, ****$P < 0.0001$). **f** Schematic diagram of nanorobot transvascular endothelial and intratumor penetration in vitro experimental models. **g** Confocal laser scanning microscopy (CLSM) shows the distribution of

UHHTN/DOX nanorobots and nanoparticles in the Transwell chamber. Scale bar, 50 µm. The left panel displays the result of transvascular extravasation, and the right panel manifests the result of intra-tumoral penetration. **h** Cross-sectional CLSM images at different thickness intervals from the top to bottom of the multicellular tumor spheroid (MTS) of about 120 µm height. Three layers represent 1/6, 1/3, and 1/2 thickness, respectively. The surface of the MTS was defined as 0 µm. Scale bar, 50 µm. **i** Quantification of the mean optical density of different layers. Data are presented as the mean ± s.d. Statistical significance was determined by one-way ANOVA with Tukey's post hoc test for comparison. ($n = 3$ replicates; **$P < 0.05$, ***$P < 0.001$, ****$P < 0.0001$). Source data for the figure are provided within the Source Data file.

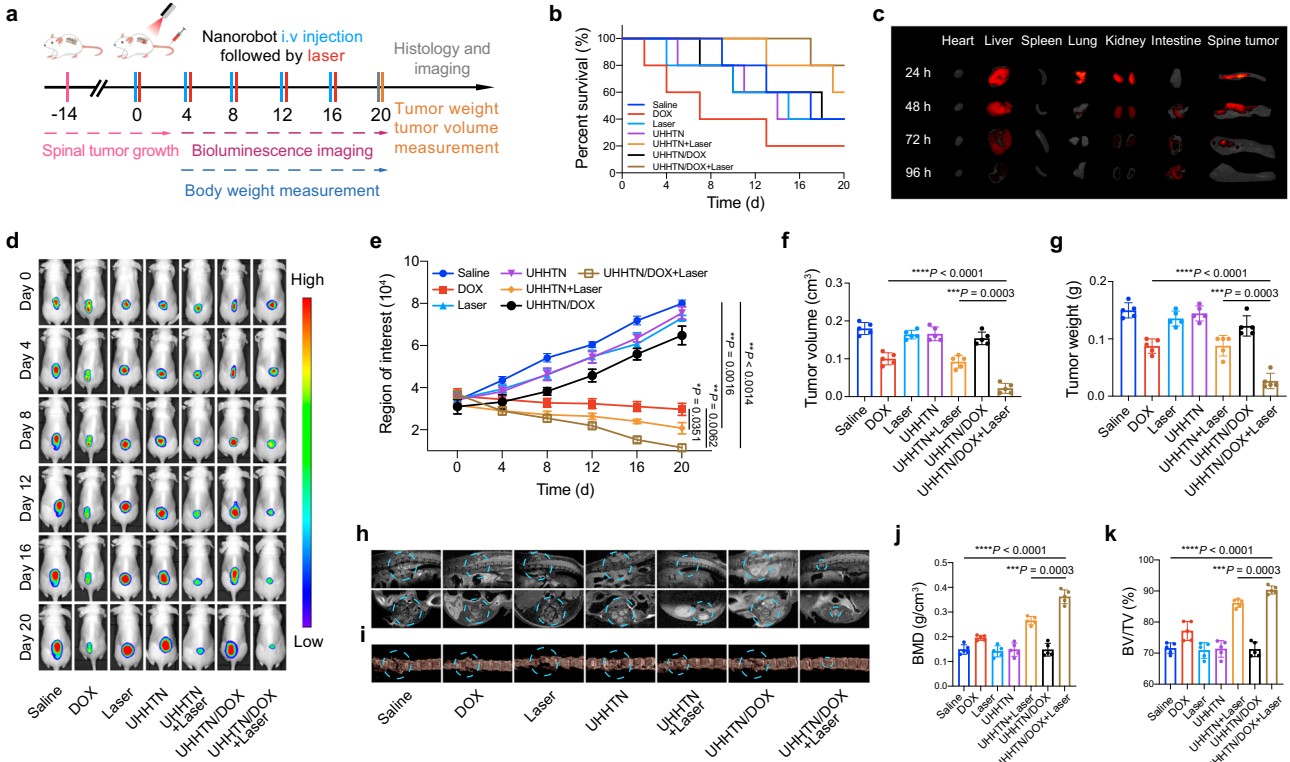

**Fig. 6 | Therapeutic efficacy of UHHTN/DOX nanorobots in the treatment of TNBC spinal metastasis. a** Timeline of different treatments and measurements in mice with TNBC spinal metastasis. **b** Survival curves of nude mice with TNBC spinal metastasis during 20 days treatment at a DOX dosage of 5 mg kg⁻¹ once every four days. **c** Distribution of UHHTN/DOX/Cy5.5 nanorobots (Dose, 5 mg kg⁻¹) at different time points in isolated spinal tumors, heart, liver, spleen, lung, and kidney. **d** Representative bioluminescence images of different groups of surviving nude mice with TNBC spinal metastasis recorded at multiple time points after receiving 20 days treatments at a DOX dosage of 5 mg kg⁻¹ once every four days. **e** Fluorescence analysis of regions of interest for TNBC spinal tumors on bioluminescence imaging at multiple time points after receiving 20 days. Data are presented as the mean ± s.d. Statistical analysis was performed by one-way ANOVA with Tukey's post hoc test. ($n = 5$ mice per group; **$P < 0.05$, ***$P < 0.001$,

****$P < 0.0001$). **f** Tumor weight and (**g**) volume in nude mice with TNBC spinal metastasis after different 20 days treatments at a DOX dosage of 5 mg kg⁻¹ once every four days. Data are presented as the mean ± s.d. Statistical analysis was performed by two-tailed Student's $t$ test. ($n = 5$ mice per group; **$P < 0.05$, ***$P < 0.001$, ****$P < 0.0001$). **h** Gross T2 MRI and **i** gross CT imaging of spines (dotted outline) in nude mice 20 days after treatment in different groups at a DOX dosage of 5 mg kg⁻¹ once every four days. **j** MicroCT bone parameters analysis of bone density and **k** bone volume/tissue volume ratio at spinal tumor lesions in different treatment groups after 20 days therapy. Data are presented as the mean ± s.d. Statistical analysis was performed by two-tailed Student's $t$ test. ($n = 5$ mice per group; **$P < 0.05$, ***$P < 0.001$, ****$P < 0.0001$). Source data for the figure are provided within the Source Data file.

treated with UHHTN/DOX+Laser were improved to 80%, compared with 60% and 40% in the UHHTN+Laser and UHHTN/DOX groups, respectively, demonstrating the synergistic effects of chemotherapy and photothermal therapy on the survival rate. In contrast, the survival rate of the DOX group was 0% (mice died on 2, 4, 7, and 13 days, respectively) due to the systemic toxic effects and adverse reactions. In an in vivo biodistribution study, mice carrying TNBC spinal tumors were collected for in vitro fluorescence imaging of spinal tumors and major organs after 24 h, 48 h, 72 h, and 96 h of intravenous injection of UHHTN/DOX nanorobots (Fig. 6c). The fluorescence intensity of spinal tumor tissue was significantly higher than that of other tissues,

indicating that the UHHTN/DOX nanorobot can passively target tumor sites. We used high performance liquid chromatography to determine the concentration of DOX in different organs to further validate the distribution of UHHTN/DOX. After 24 h of injection, the accumulation of UHHTN/DOX in the tumor increased significantly, and higher DOX was detected compared to other organs (Supplementary Fig. 49). Bioluminescence imaging and gross observation showed the progressive growth of spinal metastasis in the saline control group, indicating successful model establishment (Fig. 6d and Supplementary Fig. 50). Compared with the saline group, the UHHTN/DOX+Laser treatment significantly inhibited tumor growth, as shown by the

bioluminescence imaging, tumor weight and volume (Fig. 6e–g), suggesting the high therapeutic efficiency of UHHTN/DOX nanorobots with photothermal effects. Compared with the UHHTN+Laser and UHHTN/DOX groups, the UHHTN/DOX+Laser group had smaller tumor in the spine (Fig. 6d–g and Supplementary Figs. 50, 51). Therefore, the combination of nanorobot-mediated chemotherapy and photothermal effects was more suitable for the treatment of spinal metastasis.

To further verify whether the nanorobots can promote bone regeneration after tumor elimination. Following twenty days after injection, the MRI imaging showed that tumors with high levels of T2 signaling in the spine were obviously reduced in the UHHTN/DOX +Laser group, resulting in a nearly normal morphology in mice (Fig. 6h). Next, the bone destruction in the tumor was analyzed by 3D reconstruction of CT analysis (Fig. 6i). Osteolytic lesions in the spine could be observed in the saline and laser groups, but the internal osteolysis was significantly relieved after treatment with UHHTN/DOX nanorobots combined with NIR, indirectly suggesting the inhibitory effect of the nanorobots on the bone destruction of TNBC cells. Quantitative morphometric analysis showed that the bone mineral density (BMD) and mean bone volume/tissue volume ratio (BV/TV%) (Fig. 6j, k) in the UHHTN/DOX+Laser treatment group were 2.4 and 1.3 times higher than those in the control group, respectively. Taken together, these results demonstrated the superior efficacy of UHHTN/ DOX nanorobots in the reduction of TNBC spinal metastasis.

To confirm that the synergistic chemo- and phototherapeutic effect of UHHTN/DOX nanorobots can function in a variety of tumors with stiff stroma, we carried out in vivo anti-tumor experiments in lung cancer, pancreatic cancer, and melanoma using tail vein injection of UHHTN/DOX nanorobots. In the subcutaneous tumor models of lung cancer, pancreatic cancer and melanoma (PANC-1, A549 and B16F10) in mice, UHHTN/DOX+laser treatment significantly inhibited tumor growth and led to almost complete elimination of subcutaneous tumors, which was much better than the UHHTN+Laser and saline groups (Supplementary Figs. 52–54), suggesting that UHHTN/DOX nanorobots can be applied to the treatment of other rigid tumors.

We next explored whether UHHTN/DOX could be eliminated from the body. During one treatment cycle after UHHTN/DOX injection, urine and feces samples were collected. Si was detected in the urine and both Au and Si were detected in the feces. the possible reason is that Si in UHHTN/DOX can be excreted through the kidneys, while a small amount of Si and Au can be excreted through liver processing and bile after ingestion of the liver, and finally through the feces (Supplementary Figs. 55). To detect the long-term toxicity of Au and Si, we took different organs of mice at the end of the treatment to detect the content of Au and Si elements. It was found that a small amount of Au elements was present in the liver, spleen, and lung (Supplementary Fig. 56). Finally, hematoxylin and eosin staining (H&E) of the major organs and hematological examination indicated that our nanorobots is biocompatible and no significant side effects were shown (Supplementary Figs. 57–62).

### The effects of motion and TSM remodeling of nanorobot on extravasation and penetration

Although many studies have shown that the motility ability of nanorobots can effectively enhance tumor penetration, when applied in vivo to access tumors with stiff stroma, the dense TSM around the tumor could impede the motion of nanorobots[40,41]. It has been reported that photothermia could reduce the dense TSM; therefore, we further verified whether UHHTN nanorobots can effectively remodel TSM that further contribute to the inhibitory effect of nanorobots on tumor growth. UHHTN-mediated photothermia was obviously triggered by laser irradiation for 10 min, leading to a temperature of approximately 55.5 °C in the backs of mice (Supplementary Fig. 63).

The spinal tumors were then analyzed by histology and immunofluorescence to detect the changes in the TSM. Generally, HE staining showed that the ECM became loose with more extracellular space in the UHHTN+Laser and UHHTN/DOX+Laser groups compared to saline and the corresponding groups without laser irradiation (Fig. 7a), and meanwhile exhibited similar degradation efficacy, indicating the role of laser irradiation in regulating ECM composition. In addition, more necrotic tumor and stromal cells with karyorrhexis were observed in the nanorobots groups with laser irradiation. TUNEL staining and quantitative analysis showed that the apoptotic cells, including stromal and tumor cells with red fluorescence, were significantly enhanced in the UHHTN+Laser and UHHTN/DOX+Laser groups compared with the UHHTN and UHHTN/DOX groups (Fig. 7a, d).

Cancer-associated fibroblasts (CAFs) with α-SMA and CD31 markers and tumor-associated macrophages (TAMs) with F4/80 marker are the main stromal cells of the TSM barrier[42,43], CAFs and TAMs were abundantly disturbed in the spinal metastasis of MDA-MB-231 cells in the saline group (Fig. 7b). These cells were scarce in the UHHTN+Laser and UHHTN/DOX+Laser groups than in the UHHTN and UHHTN/DOX groups, indicating that stromal cells were significantly injured by the nanorobot-mediated hyperthermia. Importantly, the quantitative analysis showed that the MOD between the UHHTN+Laser and UHHTN/DOX+Laser groups was similar (Fig. 7e), excluding the effect of DOX on stromal cell death. The ECM surrounding stromal and tumor cells is a complex and dense network in TNBC, with collagen I and fibronectin as the dominant components[42,44]. The green fluorescence intensity representing the density of collagen I and fibronectin was significantly lower in the UHHTN+Laser and UHHTN/DOX+Laser groups than in the group without laser irradiation, demonstrating the nanorobot-mediated superior denaturation efficacy of ECM (Fig. 7c). The quantitative MOD values of collagen I and fibronectin in the UHHTN+Laser group were 52.67 and 137.2, respectively, in the UHHTN +Laser group and 44.48 and 136.6 in the UHHTN/DOX+Laser group, indicating that differences between these two groups were low (Fig. 7e); therefore the hyperthemia mediated by the nanorobots rather than DOX was the main factor for ECM degradation. The photothermal remodeling capability of nanorobots also functions in other three cellular subcutaneous tumor models in mice (A549, PANC-1 and B16F10) (Supplementary Figs. 64–66).

To prove whether TSM remodeling could open intratumoral pathways that sustain nanorobot motion in vivo and enhance accessibility to distal cancer cells for deeper tumor penetration. We next compared vascular extravasation and tumor penetration in vivo of nanorobots with TSM remodeling capability and passive core-shell structured AuNS@SiO$_2$ nanoparticles with the same TSM remodeling capability but no motion performance. The real-time extravasation and tumor penetration in subcutaneous inoculation of the MDA-MB-231 tumor model were monitored. The result shows the nanorobots were rapidly diffusion into deep tumor tissue from the vascular system and distributed throughout the observation area. The entire tumor tissue showed a nearly equal fluorescence intensity during the 120 min. In contrast, AuNS@SiO$_2$ nanoparticles were still restricted in the blood vessels at 120 min post-injection, and did not diffuse well into the tumor tissue (Fig. 7f). These results reveal that nanorobots with both motion and TSM remodeling have a superior tumor-penetration ability and can extravasate efficiently into distal tumor tissues, significantly contributing to the therapy efficacy.

## Discussion

In this work, we have demonstrated a site-selective superassembly strategy that enables the separation of a single particle surface into two regions for independent patterning. The ability to precisely control the ligand coverage location allows for one region to completely inherit the spiky surface topological structures of premade AuNSs, while the other develops into a hollow open tail, thus enabling the synthesis of

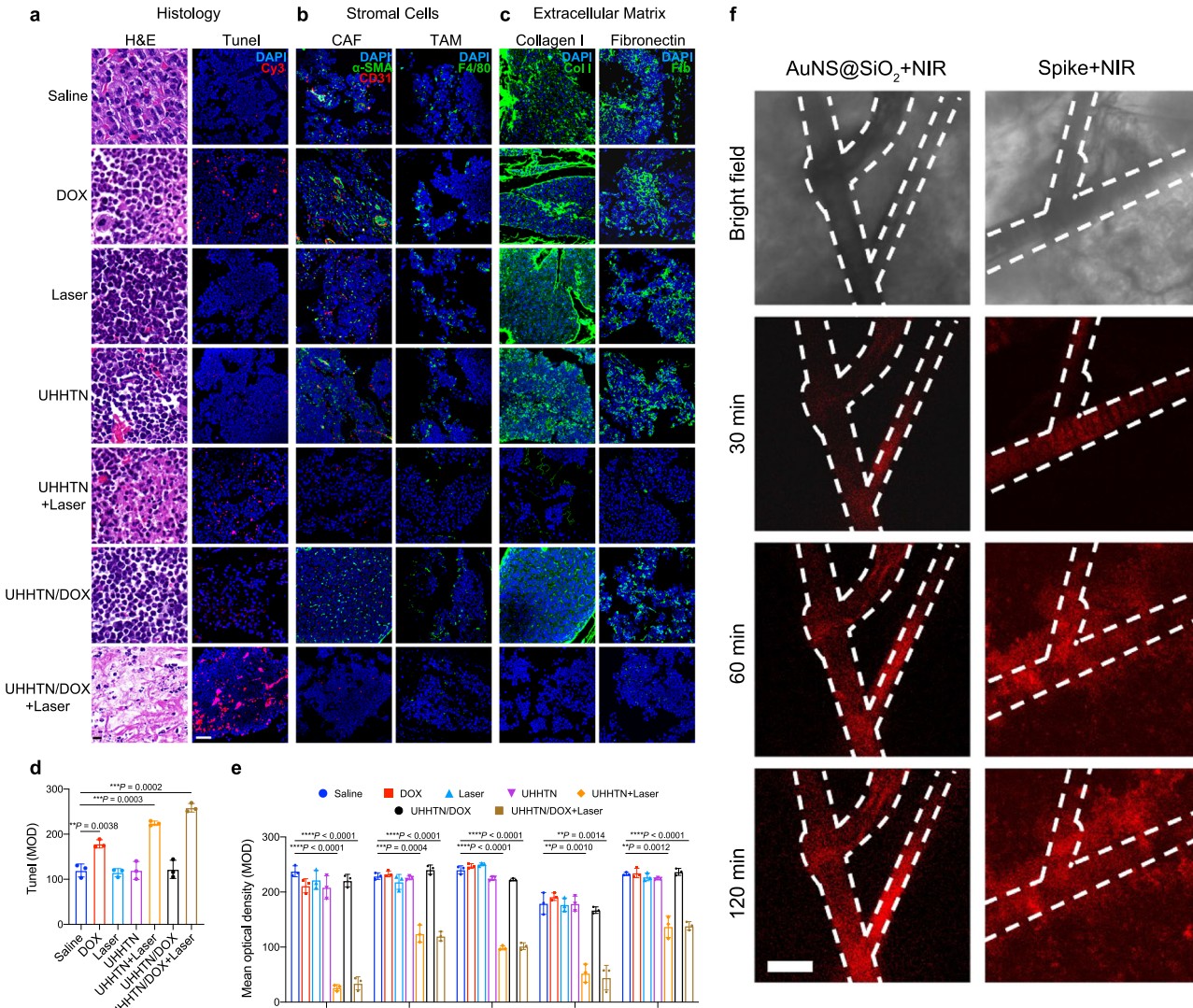

**Fig. 7 | The TSM remodeling mediated by the photothermal effects of UHHTN/ DOX nanorobots in vivo. a** HE staining and TUNEL analysis of TNBC spinal metastasis were performed 20 days after nanorobot injection at a DOX dosage of 5 mg kg⁻¹ once every four days. Scale bar, 10 µm in HE staining; Scale bar, 50 µm in TUNEL analysis). **b** Immunofluorescence showing the proportion of CAFs (α-SMA +green signal/CD31-red signal) and TAMs (F4/80 green signal) in tumors in different groups at a DOX dosage of 5 mg kg⁻¹ once every four days after 20 days therapy. Scale bar, 50 µm. α-smooth muscle actin, α-SMA; cancer associated fibroblast, CAF;Tumor-associated macrophage, TAM; 4′,6-diamidino-2-phenylindole, DAPI. **c** Immunofluorescence showing the expression of collagen I (green signal) and fibronectin (green signal) in tumors of different groups at a DOX dosage of 5 mg kg⁻¹ once every four days after 20 days therapy. Scale bar, 50 µm.

**d** Quantitative mean optical density (MOD) results from TUNEL analysis fluorescence in tumor tissues of different groups after 20 days therapy. Data are presented as the mean ± s.d. Statistical analysis was performed by two-tailed Student's $t$ test ($n = 3$ replicates; **$P < 0.05$, ***$P < 0.001$, ****$P < 0.0001$). **e** Quantitative MOD results from α-SMA, CD31, F4/80, collagen I and fibronectin fluorescence in tumors of different treatment groups after 20 days therapy. Data are presented as the mean ± s.d. Statistical analysis was performed by two-tailed Student's $t$ test ($n = 3$ replicates; **$P < 0.05$, ***$P < 0.001$, ****$P < 0.0001$). **f** Real-time microdistribution in MDA-MB-231 tumors at 30, 60, and 120 min after UHHTN/Cy5.5 nanorobot and AuNS@SiO₂/ Cy5.5 by intravenous injection at a Cy5.5-equivalent dose of 0.5 mg kg⁻¹. Scale bar, 100 µm. Source data for the figure are provided within the Source Data file.

asymmetric biomimetic urchin head/hollow tail structured NIR-triggered nanorobots. By rational head-tail design, multiple functionalities to facilitate extravasation and penetration into tumor tissue, and efficient internalization as well as efficient drug encapsulation and stimuli-responsive release capability are imparted to single nanodevices, which show highly efficient anticancer activity in multiple cancer types with stiff tumors.

The study has found that nanoparticles with multifarious topological features enable activation and amplification of immune response, our system suggest the utility of these designs could be used for cancer immunotherapy[45]. Due to remodeling tumor microenvironment capability, our system is able to solve the existing difficulty of T cells infiltrating into solid tumors for more effective

immunotherapies; In addition, this system could be used as vectors that enable encapsulation, stabilization and co-delivery of the antigens and adjuvants for cancer or antiviral vaccines[46]. The immune system activated by both spike-mediated physical cues and biological or chemical cues from loaded drugs could synergistically bolster powerful immune responses. Notably, the UHHTNs demonstrated here are based on passive targeting. Their surfaces could be further rationally engineered with functional biomolecules (e.g., targeting molecules, probes or poly-(ethylene glycol) due to their flexible surface chemistry, not only for active targeting and increased circulation time in the bloodstream for a higher accumulation efficiency in a denser tumor environment, but also enable real-time tracking and monitoring of nanorobots in living tissues so as to precisely control their behaviors

and to enhance efficacy of existing nanomedicines in numerous biological applications.

# Methods

## Materials

All chemical reagents were used in experiments without further purification. Hydrogen tetrachloroaurate (III) hydrate 99.9% (metal basis Au 49%), silver nitrate (AgNO₃), and ʟ-ascorbic acid (LAA) were supplied by Alfa Aesar; sodium citrate tribasic dihydrate (99.0%), 4-mercaptophenylacetic acid (4-MPAA, 97%), tetraethyl orthosilicate (TEOS) (≥99.0%), hexadecyltrimethylammonium bromide (CTAB), and 3-mercaptopropionic acid were purchased from Sigma-Aldrich; Poly (acrylic acid) (PAA) (average Mw = 5000) was purchased from Polymer Source, Inc. 2-Propanol (HPLC grade) and ammonium hydroxide (28.0–30.0%) were purchased from Macklin. Lauric acid (97%) and stearic acid (95%) were provided by Alfa Aesar; Cy5.5 Mono NHS Ester (Cy5.5-NHS), (3-aminopropyl) triethoxysilane (APTES), doxorubicin hydrochloride (DOX, >99%), and dimethyl sulfoxide (DMSO) were purchased by Sigma-Aldrich. Hydrogen chloride (HCl) was purchased by Aldrich Chemical Co., and 300 mesh copper specimen grids with formvar/carbon support film (referred to as TEM grids in text) were purchased from Beijing Zhongjingkeyi Technology Co. Dulbecco's modified eagle media (DMEM), fetal bovine serum (FBS), penicillin/streptomycin and phosphate-buffered saline (PBS) were purchased from HyClone (UT, USA). Four percent formaldehyde, 4,6-diamidino-2-phenylindole (DAPI), 0.1% Triton X-100, Hoechst 33342 and crystal violet were purchased from Beyotime Biotechnology (China). Cell Counting Kit-8 (CCK-8) was purchased from Dojindo (Japan). A live/dead cytotoxicity kit was purchased from KeyGEN BioTECH (China). D-luciferin was obtained by PerkinElmer (Waltham, MA, USA). Specific antibodies against a-SMA, CD31, F4/80, collagen I fibronectin and labeled secondary antibodies with label were purchased from Cell Signaling Technology (Danvers, MA, USA). Deionized water (resistance > 18.2 MΩ/cm) was used for all solution preparations.

## Synthesis of Au nanostars (AuNSs) with sharp nanospikes

In brief, 2 mL HAuCl₄ (25 mM) was added to 150 mL of H₂O in a round bottom bottle flask, followed by 200 μL of 1 M HCl and 1.45 mL of 14 nm AuNP seeds with vigorous stirring for 3 min. Then 400 μL of 10 mM AgNO₃ was injected into the reaction. Next, 1 mL of LAA (100 mM) was added rapidly. The color of the resulting solution turned green immediately. The synthesis of AuNSs with short spikes was performed through the same process except for changing the amount of LAA to 3 mL.

## Synthesis of asymmetric urchin-like head-hollow tail nanostructures (UHHTNs)

In a typical process, 4 mL of citrate-stabilized AuNS solution was centrifuged and the supernatant was removed. Then, the concentrated AuNSs were redispersed in 400 μL of deionized H₂O. The obtained AuNS aqueous solution was subsequently added dropwise to a mixture containing 2.5 mL of 2-propanol, 25 μL of 4-MPAA (25 mM in ethanol) and 25 μL of PAA (0.65 mM in H₂O) under vigorously stirring for 30 min. Afterwards, 600 μL of CTAB (0.0015 g/mL in H₂O), 2 μL of TEOS and ammonium hydroxide solution (pH=10.98) were added under mild stirring. After 6 h of reaction, the final products were isolated and collected by centrifugation at (2800 rpm, centrifugal force, 578 g) for 8 min and washed with ethanol twice for TEM/SEM characterization. The entire reaction process was performed at room temperature.

## Preparation of nanostructures with tunable spiky surface topological structures

The synthesis of nanostructure with different surface spike lengths was performed through the same processes except that the 4-MPAA concentration was adjusted to 5 mM and 15 mM, the corresponding spike length of the resultant nanostructures was -13 and 65 nm, respectively (Fig. 3).

## Cell culture

MDA-MB-231, A549, PANC-1, B16F10 and human umbilical vein cells (HUVECs) were acquired from the Institute of Biochemistry and Cell Biology, Chinese Academy of Sciences, Shanghai, China. Both types of cells were cultured in Dulbecco's modified Eagle's medium (DMEM, HyClone) containing 10% heat-inactivated fetal bovine serum (FBS, BI), and 1% penicillin/streptomycin (HyClone). All cells were incubated in a humidified incubator at 37 °C and 5% CO₂.

## Cellular uptake

MDA-MB-231, A549, PANC-1, and B16F10 cells were cultured in dishes at a density of $1 \times 10^5$ cells/well for 24 h and divided into three groups: nanoparticles with smooth heads, nanoparticles with spiky heads, and nanorobots with spiky heads under $1\,W\,cm^{-2}$ 980 mm laser irradiation. Cells were treated with 100 μg/mL nanoparticles and incubated for 0 min, 5 min, 10 min, 15 min, 20 min and 30 min or combined with laser irradiation. Then, the supernatant was removed, and the cells were washed three times with cold PBS (HyClone, UT, USA). The cells were fixed with 1 mL of 4% formaldehyde (Beyotime Biotechnology, China) per well for 10 min at room temperature, and then washed three times with cold PBS. Finally, the membrane and nuclei were stained with 1,1'-Dioctadecyl-3,3,3',3'-Tetramethylindodicarbocyanine, 4-Chlorobenzenesulfonate Salt (DID, Beyotime Biotechnology, China) and 4,6-diamidino-2-phenylindole (DAPI, Beyotime Biotechnology, China) after permeabilization with 0.1% Triton X-100 (Beyotime Biotechnology, China), and images were taken by CLSM after DAPI staining.

## Three-dimensional multicellular tumor spheroids (3D MTSs)

Three-dimensional MTSs of MDA-MB-231 cells were used to evaluate the tumor penetration ability of nanorobots. MDA-MB-231 cells with a cell density of $5 \times 10^3$ cells mL$^{-1}$ were seeded on u-type ultralow adsorption 96-well plates (Corning, USA). After 3 days of incubation, 3D MTSs were selected and treated with nanoparticles without spiky heads, nanoparticles with spiky heads, and nanorobots with spiky heads under $1\,W\,cm^{-2}$ 980 mm laser irradiation for 30 min. After Hoechst 33342 (Beyotime Biotechnology, China) staining of the cell nuclei, CLSM was used to determine the morphology of 3D MTS.

## Animal models

Six-week-old female nude mice (BALB/c) and C57BL/6 mice were purchased from Shanghai JieSiJie Laboratory Animals Co. LTD (Shanghai, China) and grown under pathogen-free conditions. All mice experiment protocols were performed in accordance with the Guidelines for the Care and Use of Laboratory Animals approved by the Animal Experimentation Ethics Committee of Zhongshan Hospital, Fudan University (2020-032). To establish a mouse model of spinal metastasis, 25 μL of MDA-MB-231 cells with luciferase labeled triple negative breast cancer (TNBC) cells (density $1 \times 10^6$ cells/ml) were injected percutaneously directly into the vertebral bodies of BALB/c nude mice. A mouse subcutaneous tumor models with different cell lines were also established, and the A549, PANC-1 and B16F10 cells were inoculated into the subcutaneous lateral abdomen of each female mouse. Nude mice with spinal tumors were divided into 7 groups (5 mice per group) as follows: Saline, DOX, Laser, UHHTN, UHHTN+Laser, UHHTN/DOX, UHHTN/DOX+Laser every 4 days (5 mg kg⁻¹; 200 μL). The irradiated group was irradiated with NIR (1.5 W cm⁻²) for 10 min. Survival curves and body weights, and tumor growth in the nude mice were monitored after 20 days treatment. At the end of treatment, the spinal and subcutaneous tumors were removed for weighing and volume measurement to evaluate the treatment effects in different groups

after anesthetization with 4% chloral hydrate injected intraperitoneally at a dosage of 0.1 mL/10 g. Mice were housed at an ambient temperature of 22 °C and 45% humidity with a diurnal cycle of 14/10 (on at 6:00 and off at 20:00). The maximum total volume of all tumors in the mice was 2000 mm³ as far as ethical norms allowed. This value was not reached during the experiment.

### In vivo real-time imaging of extravasation and tumor penetration

MDA-MB-231 cells ($1 \times 10^6$) were inoculated subcutaneously in the abdomen of female nude mice (6 weeks old) near the subcutaneous venous vessels. When the tumor reached approximately 6 mm³, the tumor-bearing mice were injected with UHHTN/Cy5.5 and AuNS@SiO₂/Cy5.5 via the tail vein for real-time observation. Then, mice were anesthetized. Without damaging the blood supply vessels, an arcuate incision was made around the subcutaneous tumor and the flap was elevated. The tumor is then exposed and fixed on a microscope slide. The coverslip is attached with enough pressure to flatten the tumor surface. Timed imaging of the tumor is performed using CLSM.

### Transvascular extravasation and intratumoral penetration

To investigate the vascular penetration of UHHTN/DOX nanorobots, HUVECs cells ($1 \times 10^5$ cells mL$^{-1}$) were seeded into 24-well Transwell inserts (polycarbonate filter, 0.4 μm pore, Corning, USA) to simulate the vascular barrier of tumor tissue. The HUVECs were treated with UHHTN/DOX nanoparticles (100 μg mL$^{-1}$) with or without 980 nm laser irradiation (1 W cm$^{-2}$) for 30 min and then incubated for another 2 h, followed by the removal of excess medium containing nanorobots, rinsing and changing the medium change. Then inserts with HUVECs were transferred to the 6-well plates with MDA-MB-231, A549, PANC-1, and B16F10 cells in the lower chamber, followed by laser irradiation or lack thereof for 30 min and coculture for 12 h. Hoechst 33342 was used to stain the inserted HUVECs cells and lower chamber MDA-MB-231, A549, PANC-1, and B16F10 cells for observation with CLSM. To explore the penetration of the UHHTN/DOX nanorobots into deep tumor cells, we used the same operation, except that the Transwell inserts were inoculated with MDA-MB-231, A549, PANC-1 and B16F10 tumor cells ($1 \times 10^5$ cells mL$^{-1}$) to mimic the external tumor cells in the tumor tissue.

### Cytocompatibility and cytotoxicity

Cell Counting Kit-8 (CCK-8, Dojindo, Japan) method was used to detect the cytocompatibility and cytotoxicity of nanorobots with different structures. To determine the cytocompatibility, MDA-MB-231, A549, PANC-1, and B16F10 cells in 96-well plates were treated with nanoparticles with smooth heads, nanoparticles with spiky heads, and nanorobots with spiky heads under 1 W cm$^{-2}$ 980 mm laser irradiation for 0–30 min. To detect the cytotoxicity, the cells were subjected to different treatments after 24 h: 1) Control, 2) 10 mg/mL DOX, 3) 1.5 W cm$^{-2}$ Laser, 4) 100 μg/mL UHHTN, 5) 100 μg/mL UHHTN + 1.5 W cm$^{-2}$ Laser, 6) 100 μg/mL UHHTN/DOX, 7) 100 μg/mL UHHTNA/DOX + 1.5 W/cm² Laser. Cells were incubated for an additional 24 h prior to CCK-8 experiments, and then 10% CCK-8 solution was added and incubated at 37 °C for 2 h. The OD values were measured using a multiplate reader (FlexStation 3, Molecular Devices, USA) based on the absorbance values at 450 nm. Cell growth viability was calculated as follows: cell viability (%) = (sample OD/control OD) × 100%.

### Live/dead cell staining assay

To further confirm the cytotoxicity of the UHHTN/DOX nanoparticles, the viability of MDA-MB-231 cells was detected after fluorescence staining with a live/dead cytotoxicity kit (KeyGEN BioTECH, China). After treatment, the MDA-MB-231 cells were incubated with PI (red for dead cells) and calcein AM (green for living cells) solution at 37 °C and

5% CO₂ for 30 min and washed with PBS, and live/dead MDA-MB-231 cells were visualized by fluorescence microscopy (BX53, Olympus).

### In vivo Imaging

Mice were anesthetized with 0.1 ml/10 g 4% chloral hydrate and injected intraperitoneally with 100 μL of d-luciferin (15 mg mL$^{-1}$; PerkinElmer, Waltham, MA, USA) per gram of body weight. Tumor growth was monitored in vivo every 4 days using a bioluminescence imaging system (Xenogen IVIS 200, PerkinElmer, MA, USA) and ex vivo images of spinal tumors, heart, liver, spleen, lung, kidney, and intestine were collected at 24, 48, 72 and 96 h post-injection (VISQUETM InVivo Elite, Vieworks). The luminescence imaging images were processed with Living Image 4.4. At the end of the experimental procedure, all mice were given free access to food and water. After completion of the treatment by nanoparticle injection and NIR radiation, MRI imaging of the spinal tumor sites in anesthetized mice was performed on a Bruker BioSpec 7T MRI scanner (Bruker, Billerica, MA, USA), and images were processed with RadiAnt DICOM Viewer software 2021.2.2. The spinal tumor sites were also analyzed using a SkyScan 1072 high-resolution μCT scanner (Bruker). After 3D reconstruction using ParaVision software 4.1 (Aartselaar SkyScan, Belgium) and NRecon software 2.1 (Aartselaar SkyScan, Belgium), the spinal tumor sites were selected as regions of interest for qualitative and quantitative analysis. The bone mineral density (BMD) and bone trabecular volume/tissue volume (BV/TV%) of the spinal tumor specimens were measured using CTAn software 1.15 (Aartselaar SkyScan, Belgium).

### In vivo metabolism

Inductively coupled plasma mass spectrometry (ICP-MS) detects the type and concentration of elements in a sample. With high specificity and low detection limits, this technique is the best method for the quantification of tissue uptake of nanobots., We injected UHHTN/DOX intravenously at a DOX dose of 5 mg/kg. To determine the excretion of UHHTN/DOX in the urine and feces, urine and feces were collected by collection tubes within 4 days after injection. The collected urine and feces were analyzed using ICP-MS to quantify silicon and gold content. To detect the long-term retainment of gold and silicon, mice were executed 20 days after injection, and heart, liver, spleen, lung, and kidney tissues were taken for gold and silicon content determination.

### Histological analysis and blood chemistry

Tumors were resected and collected to measure tumor volume and weight, embedded in paraffin, and then sectioned for further measurements. H&E-stained tissue sections were observed under a microscope, and TUNEL staining was performed to detect apoptosis in tissue sections. To further explore the nanomotor-mediated photothermal effect on remodeling TSM, the effect on CAF was first assessed by staining the sections with antibodies specific for a-SMA (CST,7817. 1:500) and CD31 (Abcam, 182981, 1:2000), respectively, followed by incubation of labeled secondary antibodies with Alexa Fluor 488-labeled (Servicebio, GB25301, 1:300) and Cy3-labeled (Servicebio, GB21303, 1:300), respectively. To assess their effect on TAM and ECM components, sections were stained with antibodies specific for F4 / 80 (CST,70076, 1:500), collagen I (CST,84336, 1:1000) and fibronectin (Abcam,2413, 1:1000), respectively, and then incubated with Alexa Fluor 488-labeled secondary antibody (Servicebio, GB25303 1:400) was incubated. As a control, cell nuclei were stained with DAPI for observation under a laser confocal microscope. Finally, to investigate the safety of UHHTN/DOX nanoparticles for in vivo intravenous injection, major tissues (heart, liver, spleen, lung, and kidney) were immediately collected for histopathological analysis after the five treatment cycles. Blood samples were immediately collected for the blood chemistry test after the five treatment cycles, and whole blood analysis include white blood cell (WBC), lymphocyte percentage (Lymph#), neutrophil percentage (Gran#), red blood cell (RBC),

hemoglobin (HGB), platelet (PLT), alanine aminotransferase (ALT), aspartate aminotransferase (AST), albumin (ALB), uric acid (UA), blood urea nitrogen (BUN), and creatinine (CR) measurements.

## Statistical analysis

We used Microsoft Excel 2021 and SPSS 25.0 (SPSS, Chicago, USA) for the statistical analysis of the collected data. All data are expressed as the mean ± standard deviation (s.d.). Two groups were compared using Student's $t$ test, and more than two groups were compared using one-way analysis of variance (ANOVA). We considered $P < 0.05$ to be statistically significant. All fluorescence quantification was created by Image J 2.3.0, and statistical graphs were created by GraphPad Prism 8.0.

## Reporting summary

Further information on research design is available in the Nature Portfolio Reporting Summary linked to this article.

## Data availability

The experimental data supporting the findings of this study are available within the article and the Supplementary Information. Additional data are available from the corresponding authors upon reasonable request. Source data are provided with this paper.

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

## Acknowledgements
This work was supported by the National Key Research and Development Program of China (2019YFC1604601 (B.K.), 2019YFC1604600 (B.K.), (2019YFC1604604) (C.Z.), the National Natural Science Foundation of China (21974029 (B.K.) and the Yiwu Research Institute Program of Fudan University (20-1-28) (B.K.), the National Natural Science Foundation of China (82172738 (J.D.), 82272457 (L.J.). We acknowledge Professor Changqi Zhao (Beijing Normal University) for the helpful technical support. Special thanks to Prof. Dongyuan Zhao (Fudan University) for his valuable suggestions and comments.

## Author contributions
M.Y., B.K. and L.J. conceived the idea of the project. M.Y., B.K. and Q.C. devised and performed the experiments and wrote the manuscript. T.L. and X.L. carried out the theoretical simulations. P.P., L.Z., S.Z. and R.Z. performed data analysis. B.K., L.J., J.W., O.T., Z.G., K.L., X.W., P.C. and J.D. supervised the research, discussed the results, and revised the manuscript. All authors discussed the results and commented on the manuscript.

## Competing interests
The authors declare no competing interests.
