## [Peer Review File · Nature Communications]

REVIEWER COMMENTS

Reviewer #1 (Remarks to the Author):

This is a revised manuscript, where the authors have somewhat tried addressing the questions. However, concerns remains, especially around rigor:

1. Method of synthesis: Under vacuum, why does the lipid and the drug enter the nanoparticle tail? Whats the driving force? For example, in Doxil prep, the doxorubicin is loaded using a charge gradient in the aqueous phase and in the fatty acid/lipid bilayer.

2. Can the authors show more representative images of the spiky nanoparticles vs control nanoparticles extravasating out? Can the tissue be sectioned and visualized using a TEM to validate that the NPs are actually leaking out and we aren't seeing an artifact?

3. On the question of why such large NPs were not present in the RES organs (typically 90% will be in these organ), now the authors respond 'In addition, it is also possible that there are differences in the amount of injected nanoparticles between different groups, as it is difficult to precisely inject nanoparticles into the tail vein of mice by a syringe for injection operation due to the very tiny tail vein, which could lead the loss of nanoparticles'. This is concerning as it points to the lack of rigor. Now suddenly, the liver and lungs are lit up, but there is still more drug in the tumor (which is hard to believe).

4. They isolated doxorubicin in organic phase (ethyl acetate). Isn't Dox water soluble, and hence will be lost in the aqueous phase?

5. Why is the concentration of Dox so high in the kidney? NPs are excreted via hepatobiliary route, and a high kidney signal means the drug is released in circulation.

6. Suppl. Fig. 36- This reviewer is not convinced that there is any difference between the NPs in terms of uptake.

7. Fig.4d. For directed motion, all the NPs need to point towards the direction of the light source when switched on. How is that achieved? For example, 2 NPs pointed in opposite direction should move away from each other when lasered. So how is directed motion achieved?

Reviewer #2 (Remarks to the Author):

The authors have extensively addressed the concerns raised by the reviewers in the previous round of review, and the manuscript has improved significantly.

Reviewer #3 (Remarks to the Author):

In this MS the authors report the synthesis and properties of a nanorobot equipped with a spiky head composed of a Au nanostar (AuNS) head half-coated with a thin layer of amorphous silica and a hollow tail connected to the AuNs for drug loading. The nanobots possess the combined abilities of active cellular internalization, efficient interfacial cellular uptake, transvascular extravasation, high drug-loading capacity, near-infrared (NIR) light-triggered release, deep tumor penetration, and photothermal therapy capacities. The authors demonstrate an impressive ability to control the complex nanobot nanostructure: the growth of the silica shell on AuNSs is well controlled and preserves the variable surface spike length (~13-94 nm) and tail length (~0-510 nm) and possesses a large tunable hollow diameter (~100-240 nm). The photothermal behavior of the AuNS remodels

the TSM by reducing stromal cell viability and causing denaturation of the extracellular matrix, creating intratumoral pathways that sustain self-propelled nanorobot motion in vivo and enhance accessibility to cancer cells. Upon NIR irradiation phase-change materials (PCMs) contained within the large hollow cavity along with (DOX) anti-cancer drugs melts enabling on-demand, triggered release. The nanobots exhibited prominent therapeutic effects, eliminating 87.78% of the tumor volume within 20 days in a spinal bone metastasis model of TNBC in nude mice.

In my original review for Nature Nanotechnology, I commented that All-in-all this work establishes an interesting new class of nanobots and successfully demonstrates their efficacy in vivo and felt the work to be sufficiently novel and significant to be considered positively for publication in Nature Nano. However I had a number of concerns that needed to be addressed to warrant publication.

1. The description/explanation of the half-coating of the AuNS with 4-MPAA and PAA is not sufficient. What causes the Janus like segregation of these ligands on the AuNS surface.
2. What templates the formation of the hollow tail? It appears from the schematic that it is a micellar like structure terminated with CTAB head groups, but how this structure forms is unclear.

3. Does CTAB form a hybrid bilayer with the tails of the PAA ligands?
4. What limits the deposition of silica to 10-nm thicknesses? And is it 10-nm on both the COOH terminated regions and CTAB terminated regions?

For points 1 -4, perhaps the schematic could be more illustrative and the synthesis approach should be more specific at the beginning as opposed to having different details be presented throughout the manuscript.

5. Measure the local thermal gradients and explain the propulsion mechanism – thermophoresis?
6. For the videos the direction and magnitude of the thermal gradient should be provided.
7. Fig 6 shows what appears to be rapid clearance of the nanobots from all the major organs plus the tumor in 24 hours yet the therapy is conducted in periods of 4 days – this doesn't make sense. How are these spiky particles cleared so rapidly as the spikes aid in their cellular internalization? Perhaps you are just showing the clearance of DOX?
8. The nanobots are pretty complex and composed of different materials. How do they degrade and get cleared from the body? Is there any hope that these could get FDA or other clinical approval?

In the revised manuscript submitted to Nature Communications, the authors addressed all my concerns to my satisfaction and I feel that this revised manuscript should be accepted in Nature Comm.

Jeff Brinker

Distinguished and Regents Professor Emeritus, Departments of Chemical and Biological Engineering and Molecular Genetics and Microbiology, UNM; Member of UNM Comprehensive Cancer Center, UNM; Fellow Emeritus Sandia National Laboratories, Distinguished Affiliate Scientist Emeritus, Sandia/Los Alamos National Laboratories Center for Integrated Nanostructures (CINT). Member: the National Academy of Engineering, the National Academy of Inventors, the American Academy of Arts and Sciences, and the National Academy of Sciences, Associate Editor ACS Nano <https://brinkerlab.unm.edu/>

Point-by-Point Reply

Manuscript ID: NCOMMS-22-45887-T

Title: Site-Selective Superassembly of Biomimetic Nanorobots Enabling Deep Penetration into Tumor with Stiff Stroma

REVIEWER COMMENTS

Responses to the reviewer's comments

All the authors would like to express our gratitude to the referees for their time and constructive comments. Our response to each comment is provided below following each comment raised.

Reviewer #1 (Comments for the Author):

Comments: *This is a revised manuscript, where the authors have somewhat tried addressing the questions. However, concerns remains, especially around rigor:*

We greatly appreciate Reviewer #1 for the valuable comments.

Comment 1. *Method of synthesis: Under vacuum, why does the lipid and the drug enter the nanoparticle tail? Whats the driving force? For example, in Doxil prep, the doxorubicin is loaded using a charge gradient in the aqueous phase and in the fatty acid/lipid bilayer.*

Response 1: Thank you for your kind suggestion. The nanoparticle tail has an open cavity. In natural condition, there is a microbubble encapsulated inside the cavity when immersed in liquid. In this case, the external solution cannot completely enter the cavity tail; however, under vacuum, the microbubble inside the cavity of the tail will be removed and become a vacuum state. Due to the air pressure, the external lipid and the drug solution will enter the hollow tail through opening. Thus, the driving force is the air pressure.

Comment 2. *Can the authors show more representative images of the spiky nanoparticles vs control nanoparticles extravasating out? Can the tissue be sectioned and visualized using a TEM to validate that the NPs are actually leaking out and we aren't seeing an artifact?*

Response 2: Thank you for your kind suggestion. In fact, we considered using tissue TEM to characterize the extravasation of our nanoparticles but failed technically. The main reasons are as follows. Although TEM has a nanoscale resolution, it is static and has a limited analysis area. From a spatial perspective, the implementation of TEM analysis is difficult. It only shows a small region of the entire tumor. In addition, the tissue sample preparation itself is random and blinded. Due to the random angle sectioning of tissue samples, it is possible that blood vessels cannot be found in the tissue. Therefore, a large number of vessels need to be searched for displaying nanorobot extravasation. On the other hand, TEM tissue analysis was performed on a Leica Ultracut

microtome cutting sections with a thickness of 60-90 nm. The size of our nanoparticles is in the range of 400-500 nm and with hard metal composition, so it is difficult to cut the whole particle completely, and the blade may be broken and damaged.

Generally, fluorescent staining of tissue sections can be used to prove the nanoparticle extravasation. To show more representative image of spiky nanoparticles vs control nanoparticle extravasation, we performed fluorescent staining of tissue sections. After the subcutaneous tumor mice were constructed, the nanoparticles were injected through the tail vein. Mice were executed after 24 h injection, and subcutaneous tumor tissues were removed, washed with PBS, and then frozen sections were prepared. The sections were incubated with FITC-conjugated CD31 antibody (Abcam, America) for 30 min at 37°C, then washed with PBS. Finally, the images were observed using CLSM. The results showed that the spike group had better extravasation than the control group.

Colocalization of nanoparticle (red) with vascular endothelial cells labelled with FITC-conjugated CD31 antibody (green) in tumour sections after tail vein injection of UHHTN/Cy5.5 nanorobot and AuNS@SiO₂/Cy5.5 (Cy5.5 equivalent dose 0.5 mg kg⁻¹). Scale bar, 20 μm. DAPI 4',6-diamidino-2-phenylindole.

Comment 3. *On the question of why such large NPs were not present in the RES organs (typically 90% will be in these organ), now the authors respond 'In addition, it is also possible that there are differences in the amount of injected nanoparticles between different groups, as it is difficult to precisely inject nanoparticles into the tail vein of mice by a syringe for injection operation due to the very tiny tail vein, which could lead the loss of nanoparticles'. This is concerning as it points to the lack of rigor. Now suddenly, the liver and lungs are lit up, but there is still more drug in the tumor (which is hard to believe).*

Response 3: Thank you very much for your professional and rigorous review, which is of great

help to improve the quality of our manuscript. We apologize for the confusion caused by that inaccurate expression. As for the low signals in liver and lung at 24 h for the first experiment, we suspect that there are some small self-nucleated silica nanoparticles due to the incomplete purification of injected UHHTN/DOX nanoparticles. The rapid clearance of small particles can contribute to the low signals in other organs 24 h after injection compared with large nanoparticles. Besides, another reason could be that the two experiments were performed on different instruments (**first**: Xenogen IVIS 200, PerkinElmer, MA, USA; **second**: VISQUETM InVivo Elite, Vieworks). In 2022, the in vivo imaging instrument for the first experiment could not be used properly because of the covid epidemic in Shanghai, China. However, for the two experiments, we can find that the fluorescence signals of nanoparticles are mainly enriched in liver, lung and kidney, showing a similar biodistribution trend in different organs.

We have added the instrument information in vivo Imaging section in **Page 38** in revised manuscript.

first:

second:

Comment 4. *They isolated doxorubicin in organic phase (ethyl acetate). Isn't Dox water soluble, and hence will be lost in the aqueous phase?*

Response 4: Thank you for your kind suggestion. To ensure the final products are only DOX-trapped UHHTN nanostructures, two rounds of washing with organic phase (DMSO) are required to remove PCM adsorbed on the outer surface of nanoparticles, unloaded fatty acids and free DOX. There are two reasons for the use of organic phase rather than aqueous phase. First, because fatty acids need to be washed off, a solvent that can dissolve fatty acids must be used. Water cannot dissolve fatty acids, so water is not used. Second, the addition of water will solidify the fatty acid.

In this case, DOX-trapped fatty acid particles will also be formed in the solution; therefore, we use organic phase to wash away the excess fatty acid that is not loaded in the cavity. After removal, water is added to solidify the fatty acid in the cavity.

Comment 5. Why is the concentration of Dox so high in the kidney? NPs are excreted via hepatobiliary route, and a high kidney signal means the drug is released in circulation.

Response 5: Generally, after i.v injection via the mice tail vein, nanoparticles will circulate throughout the body and distribute to different organs. Then, as the blood flows in body, the nanoparticles (size >5 nm) in kidney will eventually backflow to the liver and metabolized by the hepatobiliary route. Our *in vivo* imaging results also confirmed a gradual decrease in the fluorescent signal in the kidney and a gradual increase in the intestine within 96 h (**Fig 6c**). Although there was a small amount of metabolite elemental silicon in urine at 96 h due to the slight degradation of silicon particles, no metabolite elemental gold was found in urine, only in feces (**Supplementary Fig. 56**). These results suggest that the nanoparticles flow back from the kidney to the liver and are eventually metabolized mainly in liver. Also, it has been shown that nanosquares, nanotriangles and nanopentagons exhibit different levels of enrichment in the kidney (*Mol. Ther.*, 26 (3) (2018), 784-792). Therefore, at 24 h, the high concentration of DOX in kidney could be attributed to the deposition of our spike-shaped nanoparticles in the kidney rather than free dox.

Fig 6. c Distribution of UHHTN/DOX/Cy5.5 nanorobots (Dose, 5 mg kg⁻¹) at different time points in isolated spinal tumors, heart, liver, spleen, lung and kidney.

Supplementary Fig. 56. Quantification of gold and silica in urine and feces by ICP-atomic emission spectrometry after 4 days therapy (UHHTN/DOX nanorobot at a DOX dosage of 5 mg kg⁻¹). Data are presented as the mean \pm s.d. (n = 3).

Comment 6. *Suppl. Fig. 36- This reviewer is not convinced that there is any difference between the NPs in terms of uptake.*

Response 6: Differences in internalization efficacy of the three particles are mainly attributed to surface spike degrees. Besides, the hollow diameter differences in tail could further contributed the results. In the initial manuscript, we did not highlight nanostructures with different degrees of spikes to explore the effect on internalization, mainly because the change in spike degree (including number and length) is accompanied by the change in the diameter of the hollow tail and head diameter. It is difficult to independently tune the surface spike degree and hollow tail width (Fig 3). Our initial purpose is to control the growth of silica to completely inherit the spiky nanotopology of the AuNS surface in order to obtain more spikes in surface for comparing the internalization difference between the spiky surface and the smooth surface. Multifarious spiky surfaces with nanotopography, such as those of nanostars, nanopollens, and virus-like and urchin-like nanoparticles, have been recognized as powerful morphologies strongly influencing membrane binding and destabilization for improving cellular internalization (see Nat. Mater. 2009, 8, 15-23; Adv. Mater. 2013, 25, 6233-6237; Chem, 2020, 6, 1097; ACS Nano 2021, 15, 6787-6800; J. Am. Chem. Soc. 2016, 138, 6455-6462; ACS Cent. Sci. 2017, 3, 839-846; Nanoscale, 2020, 12, 14911-14918, etc.). Among the three nanoparticles with different degrees of spikes, the particle with high degree of spikes completely inherited the AuNS seed surface topology that has more spikes on the surface, and meanwhile, the hollow diameter in the tail is the smallest, so the internalization effect is the best.

Comment 7. *Fig.4d. For directed motion, all the NPs need to point towards the direction of the light source when switched on. How is that achieved? For example, 2 NPs pointed in opposite*

direction should move away from each other when lasered. So how is directed motion achieved?

Response 7: Thank you for your kind suggestion.

Macroscale (millimeter scale) directional motion such as moving away from the light source or moving towards the light source firstly requires a relatively large particle size to overcome the disturbance of Brownian motion. The smaller the particle diameter is, the greater the Brownian motion will be, which will lead to poor directionality. In this work, the size of the nanorobots used is in the range of 400-500 nm. In such scale level, the trajectories of the nanorobot are usually randomized by the thermal fluctuation. Therefore, their motion direction is relatively poor and difficult to control compared to the microrobots. In our system, the nanorobot shows a linear and directional motion at short period, whereas an enhanced Brownian motion at long period due to the random reorientations (see Nature Communications 2020, 11, 5618; J. Am. Chem. Soc. 2021, 143, 12025; J. Am. Chem. Soc., 2022, 144, 3892-3901; J. Am. Chem. Soc. 2015, 137, 4976);

Although robots in nanoscale level have poor directionality, in comparison with static nanocarriers, nanorobots with self-motile capability can undoubtedly improve the penetration efficiency. In nanomedicine, the size is particularly important, as it plays a key role in the tumor accumulation and penetration. It is known that blood vessels in tumors are characteristically 'leaky', in that they have pores that allow nanoparticles to escape and accumulate in solid tumors. The microrobots that enable controllably directional motion seem too large to pass through these pores, which may be a trade off to an effective accumulation in the tumors. In such applications, nanorobots with smaller sizes become the only viable option. Currently, it remains difficult to steer and control compared to microrobots.

Nonetheless, the control of the migration direction in microscale is meaningful. In future, additional guidance strategies, such as magnetic field assistance or topographical guidance might help to address direction issues.

Reviewer #2 (Remarks to the Author):

The authors have extensively addressed the concerns raised by the reviewers in the previous round of review, and the manuscript has improved significantly.

We appreciate Reviewer #2 for confirming the general interest and significance of our manuscript and recommending the acceptance of our manuscript.

Reviewer #3 (Remarks to the Author):

In this MS the authors report the synthesis and properties of a nanorobot equipped with a spiky head composed of a Au nanostar (AuNS) head half-coated with a thin layer of amorphous silica and a hollow tail connected to the AuNs for drug loading. The nanobots possess the combined abilities of active cellular internalization, efficient interfacial cellular uptake, transvascular extravasation, high drug-loading capacity, near-infrared (NIR) light-triggered release, deep tumor

penetration, and photothermal therapy capacities. The authors demonstrate an impressive ability to control the complex nanobot nanostructure: the growth of the silica shell on AuNSs is well controlled and preserves the variable surface spike length (~13-94 nm) and tail length (~0-510 nm) and possesses a large tunable hollow diameter (~100-240 nm). The photothermal behavior of the AuNS remodels the TSM by reducing stromal cell viability and causing denaturation of the extracellular matrix, creating intratumoral pathways that sustain self-propelled nanorobot motion in vivo and enhance accessibility to cancer cells. Upon NIR irradiation phase-change materials (PCMs) contained within the large hollow cavity along with (DOX) anti-cancer drugs melts enabling on-demand, triggered release. The nanobots exhibited prominent therapeutic effects, eliminating 87.78% of the tumor volume within 20 days in a spinal bone metastasis model of TNBC in nude mice. In my original review for Nature Nanotechnology, I commented that All-in-all this work establishes an interesting new class of nanobots and successfully demonstrates their efficacy in vivo and felt the work to be sufficiently novel and significant to be considered positively for publication in Nature Nano. However I had a number of concerns that needed to be addressed to warrant publication.

1. The description/explanation of the half-coating of the AuNS with 4-MPAA and PAA is not sufficient. What causes the Janus like segregation of these ligands on the AuNS surface.
2. What templates the formation of the hollow tail? It appears from the schematic that it is a micellar like structure terminated with CTAB head groups, but how this structure forms is unclear.
3. Does CTAB form a hybrid bilayer with the tails of the PAA ligands?
4. What limits the deposition of silica to 10-nm thicknesses? And is it 10-nm on both the COOH terminated regions and CTAB terminated regions? For points 1-4, perhaps the schematic could be more illustrative and the synthesis approach should be more specific at the beginning as opposed to having different details be presented throughout the manuscript.
5. Measure the local thermal gradients and explain the propulsion mechanism – thermophoresis?
6. For the videos the direction and magnitude of the thermal gradient should be provided.
7. Fig 6 shows what appears to be rapid clearance of the nanobots from all the major organs plus the tumor in 24 hours yet the therapy is conducted in periods of 4 days-this doesn't make sense. How are these spiky particles cleared so rapidly as the spikes aid in their cellular internalization? Perhaps you are just showing the clearance of DOX?
8. The nanobots are pretty complex and composed of different materials. How do they degrade and get cleared from the body? Is there any hope that these could get FDA or other clinical approval?

In the revised manuscript submitted to Nature Communications, the authors addressed all my concerns to my satisfaction and I feel that this revised manuscript should be accepted in Nature Comm.

We appreciate Reviewer #3 for the high recognition of the significance and novelty of our works and recommending the acceptance of our manuscript.

REVIEWER COMMENTS

Reviewer #1 (Remarks to the Author):

The authors have addressed some of the concerns. However, some concerns still remain unaddressed:

1. The whole premise of the paper is based on the fact that these robots move and hence they promote tumor penetration. All nanoparticles move. Its supposedly directed motion that should increase tumor penetration. But now the authors reply 'Although robots in nanoscale level have poor directionality, in comparison with static nanocarriers, nanorobots with self-motile capability can undoubtedly improve the penetration efficiency'. Given that it is motility that is being highlighted as the key point of this paper, this disconnect with what exactly drives penetration weakens the paper. All nanoparticles show enhanced motion when in solution and lit up!

2. 400-500 nm particles are large particles. Such particles are rarely able to evade the RES system, hence the lungs and liver are lit up. Can the authors actually measure the fraction of total doxorubicin in the tumor vs Doxil. If this is ever to be translated, it needs to be better than Doxil.

Point-by-Point Reply

Manuscript ID: NCOMMS-22-45887-T

Title: Site-Selective Superassembly of Biomimetic Nanorobots Enabling Deep Penetration into Tumor with Stiff Stroma

Reviewer #1 (Remarks to the Author):

Comments: *The authors have addressed some of the concerns. However, some concerns still remain unaddressed:*

We thank the reviewer again for the valuable comments.

Comment 1. *The whole premise of the paper is based on the fact that these robots move and hence they promote tumor penetration. All nanoparticles move. Its supposedly directed motion that should increase tumor penetration. But now the authors reply 'Although robots in nanoscale level have poor directionality, in comparison with static nanocarriers, nanorobots with self-motile capability can undoubtedly improve the penetration efficiency'. Given that it is motility that is being highlighted as the key point of this paper, this disconnect with what exactly drives penetration weakens the paper. All nanoparticles show enhanced motion when in solution and lit up!*

Response 1: Thanks for the reviewer's comment.

When nanoparticles were injected via the tail vein, they would accumulate into tumor tissue based on EPR effect. Then, upon NIR-light irradiation, the accumulated nanoparticles will generate forces that lead to their enhanced motion. Although such motion is not directional, autonomous and active motion promoted the diffusion of nanoparticles inside tumor, leading to enhanced penetration depth, which is consistent with our experimental results of multicellular tumor spheroid penetration as shown in Fig 5h. Furthermore, self-propelled nanorobots that facilitate deep tissue penetration has been widely accepted and verified by many studies at two-dimensional planar cellular level (e.g., cellular uptake), three-dimensional multicellular spheroids (e.g., cellular penetration) (*J. Am. Chem. Soc.*, 2022, 144, 3892-3901; *J. Am. Chem. Soc.* 2015, 137, 4976, etc.), and even at *in vivo* level (e.g., tumor tissue penetration) (*J. Am. Chem. Soc.*, 2021 143, 12025; *Nature Communications* 2020, 11, 5618; *Angew. Chem. Int. Ed.* 2020, 59, 14458, etc.).

Comment 2. *400-500 nm particles are large particles. Such particles are rarely able to evade the RES system, hence the lungs and liver are lit up. Can the authors actually measure the fraction of total doxorubicin in the tumor vs Doxil. If this is ever to be translated, it needs to be better than Doxil.*

Response 2: Thanks for the reviewer's suggestion.

Although large particles are more easily to be trapped by RES than small nanoparticles (<100 nm), some studies proved that nanoparticles around 300 nm (*Angew. Chem. Int. Ed.* 2017, 56, 8446;

J. Mater. Chem., 2009, 19, 4102–4107; *Adv. Mater.* 2016, 28, 1963, etc.) and even 400 nm (*Chem. Rev.* 2021, 121, 1746; *Nat. Biomed. Eng.*, 2019, 3, 729) still can go through vascular gaps and accumulate in tumor. Besides, tumors often have a discontinuous endothelial layer depending on the size of the gap in the vessel wall, and fenestrations can range from 50 nm to 4,700 nm (*Int. J. Nanomed.* 2014, 9, 2539). On the other hand, compared to spherical nanoparticles, rod-like or disc-shaped nanoparticles can accumulate into tumors faster (*Angew. Chem. Int. Ed.* 2011, 50, 11417) and with more retention (*Adv. Mater.* 2008, 20, 1630; *Nanomedicine: Nanotechnol. Biol. Med.* 2013, 9, 686), apparently because of enhanced penetration through the pores as a result of the shortest dimension of the nanoparticle. Therefore, in this study, our nanoparticles are rod-like shape with ~285-nm head in width and ~351-nm body in length, which can accumulate into tumors by EPR effects.

The key advancement of this study is the development of a new site-selective assembly strategy and fabrication of unique asymmetric biomimetic urchin-like head/hollow tail nanorobots, which have never been achieved before. Such control capability of site-selective growth is a significant advance in nanoscience. These rationally designed nanorobots integrates multiple functionalities including active motion, TSM remodeling and enhanced nano-bio interactions as well as efficient drug encapsulation and stimuli-responsive release capability into a single nanocarrier, which have great potential for efficient cancer therapy. Thus, we finally verified these properties by *in vitro* and *in vivo* experiments. These functionalities do provide efficient therapeutic outcomes and emphasize the novel properties of our material system. However, there are large difference in physicochemical properties between inorganic nanoparticles and lipid-based Doxil, including composition, size, morphology, and surface charge and properties, etc. Therefore, direct comparison of the fraction of total doxorubicin in the tumor is not the focus of this study and lacks rigor. But we really appreciate your suggestion, and in further study we would systematically study the clinical translation potential of our nanoparticles and compare the therapeutic outcomes with clinically approved drugs.